

# Evaluating the carbon sequestration potential of volcanic soils in South Iceland after birch afforestation

Matthias Hunziker[1], Olafur Arnalds[2], Nikolaus J. Kuhn[1]

[1] Department of Environmental Sciences,
Physical Geography and Environmental Change,
Klingelbergstrasse 27,
CH-4056 Basel

[2] Faculty of Agricultural and Environmental Sciences,
Agricultural University of Iceland,
Hvanneyri,
Iceland

Correspondence to: Matthias Hunziker (matthias.hunziker@unibas.ch)

e-mail: matthias.hunziker@unibas.ch

phone: +41 61 2073645 / fax: +41 61 2070740

**Abstract.** Afforestation is a strategy to sequester atmospheric carbon in the terrestrial system and to enhance ecosystem services. Iceland's large areas of formerly vegetated and now degraded ecosystems therefore have a high potential to act as carbon sinks. Consequently, the ecological restoration of these landscape systems is part of climate mitigation programs supported by the Icelandic government. The aim of this study was to explore the change of the soil organic carbon (SOC) pools and to estimate the SOC sequestration potential during the re-establishment of birch forest on severely degraded land. Differently aged afforested mountain birch sites (15, 20, 25 and 50 years) were compared with sites of severely degraded land, naturally growing remnants of mountain birch woodland and grasslands which were re-vegetated using fertilizer and grass seeds 50 years ago. The soil was sampled to estimate the SOC stocks and for physical fractionation to characterize the quality of the SOC.

The results of our study show that the severely degraded soils can potentially sequester an additional 20 t C ha$^{-1}$ (0-30 cm) to reach the SOC stock of naturally growing birch woodlands. After 50 years of birch growth, the SOC stock is lower than that of naturally growing birch woodland. Hence, afforested stands can sequester additional SOC after 50 years of birch growth. The SOC fractionation revealed that at all tested sites most of the carbon was stored in the < 63 μm fraction. However, the particulate organic matter (POM) fraction was enriched most during the succession of afforested mountain birch stands (+ 12 t POM-C ha$^{-1}$). The study also found a doubling of the dissolved organic carbon (DOC) concentration after 50 years of birch growth. Therefore, we assume that carbon deriving from the afforestation process is sequestered as labile SOC, which may



be partly released to the atmosphere during the process of stabilization with the mineral soil phases in the future. Our results are limited in their scope since the selected sites do not fully reflect the heterogeneity of landscape evolution and the range of soil degradation conditions. As an alternative, we suggest using repeated plot measurements instead of space-for-time substitution approaches for testing C changes in severely degraded volcanic soils. Our findings clearly show that detailed

measurements on the SOC quality are needed to estimate the SOC sequestration potential of restoration activities on severely degraded volcanic soils is needed, rather than only measuring SOC-concentration and SOC stocks.

**Keywords**: land restoration, afforestation, sequestration potential, SOC stock, SOC quality, soil fractionation, Andosols

# 1 Introduction

## 1.1 Iceland's soil carbon sequestration potential by land restoration

The Icelandic government approved activities including revegetation and afforestation in the 1990's to increase the terrestrial carbon sequestration from the atmosphere (Sigurdsson and Snorrason, 2000; Aradottir and Arnalds, 2001; Ministry for the Environment, 2007). In effect, land reclamation has been carried out for over 100 years, in order to halt land degradation and soil erosion events (Crofts, 2011) caused by human activities since the islands settlement about 1100 years ago (Aradottir

and Arnalds, 2001), as well as natural stress factors, such as volcanic eruptions or the harsh climate.

Woodlands and species-rich heathlands form the un-disturbed ecosystem type on drylands at lower elevation (< 400 m asl) (Aradòttir et al., 1992). Fertile Brown Andosol is the typical soil type of these ecosystems and it is found across 13,360 km$^2$ (Óskarsson et al., 2004). Andosols have a tendency to accumulate higher quantities of SOC than other soil types, due to the cover of soil organic carbon (SOC) enriched surface horizons by volcanic ejecta and the andic properties resulting from the

formation of organo-mineral complexes (Dahlgren et al., 2004; McDaniel et al., 2012; Delmelle et al., 2015; Arnalds, 2015a). Hence, the average SOC stock of the Brown Andosols is estimated at 227 t C ha$^{-1}$ (Óskarsson et al., 2004). During the last centuries, about 43,000 km$^2$ of Iceland (~40%) have been affected by severe extreme soil erosion (Arnalds et al., 2016). Consequently, about 120-500 Mt of SOC have been lost in the past (Óskarsson et al., 2004). Presently, approximately 45,000 km$^2$ (~45 %) of the land area is covered by sparely vegetated areas which range to barren deserts, in addition to

disturbed areas with reduced carbon levels (Arnalds, 2015b). These landscapes are characterized by limited vegetation cover on vitric soil types (Arnalds et al., 2013) with low biomass production and low SOC stocks (Óskarsson et al., 2004). Vitrisols (Vitric Andosols and Leptosols), which are the typical soil types of the deserts, contain less than 45 t C ha$^{-1}$ on average (Óskarsson et al., 2004).

Based on these differences, the potential of the severely degraded soils to sequester high amounts of carbon has been

demonstrated (Arnalds et al., 2000; Ágústsdóttir, 2004). An important aspect of reclaiming degraded land is the recovery of ecosystem services including rehabilitation of farm land, protection against soil erosion or public recreation (Aradóttir et al., 2013). For example, the large-scale project called *Hekluskógar* was established in South Iceland in 2007 with the aim of



restoring resilient birch woodlands on about 900 km$^2$ in the vicinity of Mount Hekla, in order to reduce the effect of volcanic hazards (Aradóttir, 2007).

## 1.3 Assessment of SOC change in Iceland

The Icelandic carbon stocks have been reported in a national inventory for the UNFCCC (Hellsing et al., 2016). The

Icelandic National Inventory Report uses a country specific soil carbon sequestration factor of 0.51 t C ha$^{-1}$ yr$^{-1}$ for soils during the conversion of severely degraded land ("Other Land") to forest land or grassland (Hellsing et al., 2016). This is based on Icelandic field studies which found an increase in the SOC stock and therefore assigned a positive soil carbon sequestration effect to the reclamation (Aradóttir et al., 2000; Snorrason et al., 2002; Ritter, 2007; Bjarnadottir, 2009; Kolka-Jónsson, 2011; Arnalds et al., 2013). In addition, the Icelandic Soil Conservation Service continuously reviews this value by

ongoing C sequestration monitoring (Hellsing et al., 2016). The establishment of a vegetation community passes through different development stages, consequently, the sequestration rate is not linear until the new SOC stock equilibrium is reached (Smith et al., 1997; Six et al., 2002; Stewart et al., 2007). Hence, the development of the SOC stock and the SOC sequestration rates need to be recorded with a high temporal resolution instead of using the data of only two inventories (e.g. $t_0$ and $t_1$ or initial and final vegetation type).

Monitoring the total C stock is not sufficient to characterize the overall potential for removal of atmospheric C by afforestation since soil organic matter (SOM) consists of a heterogeneous mixture with respect its physical protection and chemical structure (Schmidt et al., 2011). This leads to dynamic patterns of SOC stocks, composition of SOC, decomposability and turnover rates of SOC during land-use changes (von Lützow et al., 2008; Poeplau and Don, 2013). Recent studies show that the labile SOC pool, which is composed mainly of particulate organic matter (POM), increases

simultaneously with the total SOC in the mineral soil during the establishment of vegetation systems with a higher net primary production rate (Guidi et al., 2014; Gabarrón-Galeote et al., 2015; Trigalet et al., 2016; Hunziker et al., 2017). To date, however, soil studies in Iceland have not focused on such changes of SOC fractions during the establishment of woody vegetation systems.

In Iceland, conversion of vegetation cover is currently driven by revegetation of severely degraded land (Aradóttir et al.,

2000; Arnalds et al., 2013), natural succession following glacier retreat (Vilmundardóttir et al., 2015) and by afforestation of different types of tree species on heathland (Ritter, 2007), as well as on grazed land (Snorrason et al., 2002). However, there is limited information concerning carbon sequestration in soils associated with the afforestation of severely degraded landscapes by the only native forest tree species, *Betula pubescens* Ehrh. *ssp. czerepanovii*.

The present study was part of the *CarbBirch* project (Halldórsson et al., 2011) which was launched in 2008 and involved two

of the five *CarbBirch* study areas. The main goal of *CarbBirch* was to study the ecological impact of the restoration activities in *Hekluskógar*. The present study aims at characterizing the long-term carbon sequestration potential of afforestation efforts with mountain birch on severely degraded soils. For this, we compared the SOC patterns in mountain birch stands of different ages to those of severely degraded and barren areas, reclaimed grasslands and natural old growth



birch woodlands. The article first introduces the common used SOC parameters: SOC concentration and the SOC stocks (0-30 cm), then discusses the vertical distribution of SOC, SOC quality and the interaction between the SOC and the volcanic clay minerals.

## 2 Material and methods

### 2.1 Study approach

The study area is in the vicinity of the Mount Hekla volcano (Figure 1; A). Due to the unsustainable land use and volcanic activity, most of this area has been affected by erosion. The resulting landscape is characterized by sandy deserts (Arnalds et al., 2016), which often leads to the formation of important sandstorms (Crofts, 2011; Arnalds et al., 2016), and in consequence, reclamation activities have been carried out over the last decades (Halldórsson et al., 2011). The soil parent

material generally consists of lava field material, glacial till, aeolian deposits or buried soil materials (Dugmore et al., 2009; Thorarinsdottir and Arnalds, 2012).

The afforested woodland area "*Gunnlaugsskógur*" is located approximately 1 km north of the Icelandic Soil Conservation Service Headquarters at Gunnarsholt (Figure 1; C). In 1926, the eroded area was excluded from sheep grazing by fencing. After the stabilization of the ground surface and the fertilization of the soil, birch was seeded on small plots in 1939 and

1945. In 1945, birch seedlings resulting from the activity in 1939 were transplanted on a nearby lava field, although most of the present birch area at *Gunnlaugsskógur* has naturally regenerated through seed production of the previously planted birches (Aradóttir, 1991; Aradottir and Arnalds, 2001). The age of the afforested birch sites was determined by dendrochronology as part of the *CarbBirch* project. The mean ages of the sampled afforested birch plots were 15, 20, 25 and 50 years (Birch15, Birch20, Birch25, Birch50), respectively. In addition to the birch plots, soil samples were taken from

three severely degraded sites with barren surfaces, and from three revegetated sites with grass vegetation north of *Gunnlaugsskógur* (Figure 1; D). In the present study, the severely degraded and eroded sites (Barren Land) represent the stage before any restoration activity has begun. The Barren Land sites were selected at 4 km distance from *Gunnlaugsskógur*, as barren areas were not available near the afforested birch sites, and it was assumed that the geologic and pedologic characteristics were comparable to those at the birch sites. The grassland sites (Grass50) were located next to the

severely degraded sites; these were protected against sheep grazing by fencing and then revegetated by using fertilizers and grass seeds about 50 years ago and at present are not used for hay production. The topsoil at these sites has been found to be degraded, while horizons buried by wind deposits may contain some carbon (Arnalds, 2010; Arnalds et al., 2013). This has to be considered when assessing carbon sequestration by actual restoration programs because parts of the found SOC may resulting from earlier vegetation. Due to the same age of Birch50 and Grass50, the two different reclamation types can be

compared directly. The differently aged birch sites were further compared to a naturally growing birch woodland located at "*Hraunteigur*" (Figure 1; B). This area was protected against sand encroachment by two streams but was subjected to deposition of large amounts of dust and periodic tephra fallout. Thus, it represents the original mountain birch woodlands



(Birchnat), which covered large areas in the vicinity of Mount Hekla in the past (Árnason, 1958). As the vegetation cover is subject to large scale sediment deposition, the area has accumulated soils with depth of more than 2 m (Kolka-Jónsson, 2011).

<< Figure 1 HERE >>

Field sampling was carried out in summer 2011. Each of the land cover types and age categories described above was represented by three test sites, resulting in a total of 21 sampling sites (Figure 1; E). After removing the litter layer, the top 30 cm of the mineral soil was sampled. This sampling depth interval represents the common depth for SOC stock inventories
(Aalde et al., 2006; Snorrason, 2010), and in addtion, the top 30 cm of the mineral soil contains most of the belowground living root biomass at grassland and birch sites (Snorrason et al., 2002; Bjarnadottir et al., 2007; Hunziker et al., 2014). Thus, the dominant belowground organic carbon source is located between 0 and 30 cm soil depth.

At each site, five soil pits were randomly placed. At the woody sites, sampling occurred within one half of the crown diameter of a dominant mountain birch (*Betula pubescens* Ehrh. *ssp. czerepanovii*) tree. The soil was sampled with a
cylindric metal core (Eijkelkamp Soil & Water, Giesbeek) of 100 cm$^3$ volume and 5 cm in diameter at given soil intervals (0-5, 5-10, 10-20 and 20-30 cm). The five sub-samples per depth interval were immediately mixed in order to form one composite sample. Thus, each depth interval per category was represented by three composite samples (Figure 1), resulting in a total of 84 composite samples.

### 2.2 Laboratory soil treatments

### 2.2.1 Determining common properties for volcanic soils

All 84 composite soil samples were dried at 40 °C until a constant weight was reached. The weight [g], the volume [cm$^3$] and the bulk density [g cm$^{-3}$] of the fine earth (< 2 mm) were determined by dry sieving and water displacement of the coarse material (> 2mm). Soil reaction (pH value, [-]) was determined in water (1:2.5) and potassium chloride (1:2.5 0.01 M KCl) to determine the protons in the actual and potential liquid soil phase (FAL, 1996). Acid ammonium oxalate extractable Al,
Fe and Si and pyrophosphate extractable Al and Fe were measured with an ICP device after the method of Blakemore et al., (1987). The concentrations [%] of the volcanic clay minerals allophane and ferrihydrite were estimated by multiplying the $Si_{ox}$ concentration by 6 and the $Fe_{ox}$ concentration by 1.7, respectively (Parfitt and Childs, 1988; Parfitt, 1990). The allophane and ferrihydrite concents were then summed up to determine the clay content [%] deriving from the oxalate extraction which is a typical measure for texture analysis in volcanic soils (Arnalds, 2015a).





### 2.2.2 Soil and soil organic carbon fractionation

A commonly used method for SOC fractionation is the one developed by Zimmermann et al., (2007) which produces four different functional carbon groups due to the expected reactivity of the SOC within the groups. We slightly modified the separation procedure limiting the analysis to disaggregation, the particle size separation and the density fractionation to

separate the SOC. The applied physical fractionation technique is suited to investigate the responses of SOC stability to land-use changes (Cambardella and Elliott, 1992; Six et al., 1998; Poeplau and Don, 2013, Hunziker et al., 2017).

The fractionation procedure determined the fine soil fraction (< 2 mm) of the 180 composite samples. Initially, the samples were dispersed by an ultrasound treatment (22 J ml$^{-1}$) in 150 ml deionized water to retrieve only primary organo-mineral complexes (Christensen, 2001). The samples were subsequently wet sieved to 63 microns to separate the stable sand-sized

aggregates and the un-protected particulate organic matter from the material < 63 microns. The particulate organic material (POM) was separated from the denser organic material in the mineral-associated sand and aggregate fraction (heavy fraction; HF) by density fractionation (1.8 g cm$^{-3}$, SPT from Sometu) on the soil material (> 63 microns). After separation, both fractions were washed with deionized water until the electrical conductivity of the rinse water reached < 50 µS (Wagai et al., 2008).

In some cases, the pumice material around Mount Hekla has a density of about 1 g cm$^{-3}$ (Arnalds, 2000), and some POM samples were contaminated with pumice material. We solved this problem by using a charged glass surface to separate the POM material from the pumice material (Kaiser et al., 2009). The electrostatically charged glass plate was set 2 to 5 cm above the stone plate on which the contaminated POM fraction was distributed and was slowly moved over the sample. The distance between the charged glass surface and soil particle surface was manually set due to the different sizes of POM and

pumice material. The organic particles electrostatically attracted to the glass plate were visually checked for possible "contamination" by pumice material. In these cases, the pumice material was manually removed. The pumice material (< 1.8 g cm$^{-3}$) was transferred to the HF fraction.

The material which is smaller than 63 microns represents the SOC pool of the silt and clay size fraction which can also contain aggregates consisting of volcanic clay minerals. Further, after settling time, a sample of the suspension (< 63

microns) was taken, filtered with a 0.45 microns filter and analyzed for its dissolved organic carbon content (DOC). The value of the DOC concentration was used as an indicator of the ability of the sampled soils to leach dissolved organic carbon. Compared to Zimmermann et al., (2007), the present study did not conduct oxidation with sodium hypochlorite (NaOCl) to determine the resistant SOC pool.

All samples of the bulk soil (< 2 mm) and the POM, HF and < 63 µm fractions were ball-milled and analyzed for organic

carbon content [%] by dry combustion (Leco CN 628 Elemental Determinator). The DOC content [mg l$^{-1}$] was measured using a combustion analytic oxidation method (TOC-5000A, Shimadzu).



## 2.3 SOC stock estimation

The amount of soil organic carbon that is stored in a given soil profile is defined as the SOC stock and is given in tons per hectare. According to Ellert et al., (2008) and Rodeghiero et al., (2009), the SOC stock ($SOC_{stock}$; [t C ha$^{-1}$]) is a function of the soil's carbon content ($SOC_{conc}$; [mg g$^{-1}$]), the bulk density ($BD_{<2mm}$; [g cm$^{-3}$]) of the fine soil fraction (< 2 mm) and the

investigated soil depth (d; [cm]). The conversion factor between the units is 100. Hence, the study calculated the amount of soil organic carbon which was stored in the fine soil fraction within the top 30 cm.

$$SOC_{stock} = SOC_{conc} \times BD_{<2mm} \times d \times 100$$

SOC stocks stored in the SOC fractions were calculated after Poeplau and Don (2013) and Guidi et al., (2014).

## 3 Results and Discussion

### 3.1 Physical, chemical and morphological characteristics of the sampled soil intervals

The soil material sampled at all depth intervals were andic ($(Al + \frac{1}{2}Fe)_{ox} > 2$ %) and therefore classified as Andosols (Table 2) (IUSS Working Group WRB 2014). According to the Icelandic soil classification (Arnalds, 2008), freely-drained soils under vegetation are termed Andosols, and desert soils are classified as Vitrisols. The results of the present study confirmed

that the sampled soils of the birch and grass stands are classified as Brown Andosols (1-12% C and > 6 % allophane) (Table 1, Table 2). The calculated bulk densities of the fine earth material fractions were within the range of 0.3 to 0.8 g cm$^{-3}$ of typical Icelandic Andosols (Arnalds, 2008). The severely degraded and un-vegetated soils of Barren Land were classified as Vitrisols (Arnalds, 2015c) due to the relatively high pH-values (> 7.0; H$_2$O). The pH-values of these soils were significantly

higher compared to those of the other tested land cover categories. However, the soils of Barren Land contained more organic carbon than usually found in Vitrisols of Icelandic deserts (Table 1) (Óskarsson et al., 2004). The high carbon and clay content found on Barren Land are more representative of severely degraded soils than Icelandic desert soils. This is highlighted by their high clay contents, which comprises a mixture of allophane and ferrihydrite minerals, both of which were found at Barren Land and Grass 50 sites (Table 2). These high concentrations stand in contrast to the typically low

concentration (2-5 %) found in desert Vitrisol (Arnalds, 2015d). The high ferrihydrite contents at Barren Land can be attributed to subsurface horizons (Arnalds, 2015d). Based on these findings, our results indicate that the soils tested as part of the present study are pedogenetically developed. Compared to the concentration of the ferrihydrite clays, the allophane clay minerals predominated in all samples (Table 2). These findings must be kept in mind during the evaluation of carbon sequestration in degraded soils.





Distinct trends were also seen in nutrient contents, for instance, the SOC concentration varied between 0.6 and 9.8 % within the whole dataset of the 84 samples (Table 1). Surprisingly, the lowest C contents were not found at the Barren Land sites as was expected; this is most likely due to the absent vegetation cover at the time of sampling. This finding supports the hypothesis that organic carbon was sequestered in the soil before the onset of soil erosion. Further, at the Barren Land,

Grass50 and Birch15 sites, the SOC concentrations were higher in the deeper sampling intervals (5-30 cm) than in the shallow depths of 0-5 cm (Table 1). At Barren Land, Grass 50 and the afforested Birch sites, the C:N ratios of the soil mostly varied between 10 and 14 [-]. However, the ratio was considerably higher in the top 5 cm at the birch and grass sites, compared to the soil at Barren Land or in deeper soil layers (Table 1). This pattern was most distinct in the soils at Birchnat. Hence, the soils at Birchnat also showed the highest C:N ratios with a maximum value of 19.2 (Table 1). These findings can

be attributed to the presence of freshly deposited and less decomposed organic matter close to the surface ground at the vegetated sites. In contrast, the C:N ratio was slightly higher at deeper sampling intervals at Barren land which gives evidence that the carbon originates from past vegetation cover.

The C analysis at these sites showed that un-vegetated, severely degraded volcanic soils contained appreciable amounts of SOC and that afforestation with mountain birch increased the soil C concentration during the first 50 years of shrub

establishment, predominantly in the top 10 cm. Further, the values of bulk density and SOC concentration are inversely proportional. Consequently, to further discuss the SOC sequestration potential of the soils studied, further detailed information on SOC stocks and SOC quality are needed, in addition to measurements of SOC concentration. Lastly, the unknown influence of the sampling depth needs also to be accounted for.

<< Table 1 HERE>>

<< Table 2 HERE>>

### 3.2 Studying the usually tested bulk SOC stocks

The present study found a continuous increase in median SOC stock (0-30 cm) with birch stand age (Birch15: 31; Birch20: 33; Birch25: 36; Birch50: 46 t C ha$^{-1}$) (Figure 2). During the period of 35 years (Birch15-Birch50), the sequestration rate is 0.42 t C ha$^{-1}$ a$^{-1}$ on average, without taking the SOC stock of Barren Land (as status before afforestation begins) as reference for calculation. The rate is lower than the given removal factor of 0.51 t C ha$^{-1}$ a$^{-1}$ for afforestation activities (Hellsing et al., 2016). Due to the lower SOC stock (0-30 cm) at Birch15, Birch20 and Birch25 than at Barren Land, the given results of the

SOC stocks (0-30 cm) might lead to the assumption that the soil acts as C source during the first 25 years of the establishment of birch. This would be in accordance with Hunziker et al., (2017) who found a decline of the SOC stock (0-30 cm) during the first 40 years of green alder encroachment on former subalpine pastures. Another finding of the present study is that after 50 years of birch growth, the SOC stock was still considerably lower than that of the old-growth woodlands of



Birchnat ($\Delta$ 15 t ha$^{-1}$) (Figure 2). This means that the soils at Birch50 can sequester additional organic carbon during the succession towards mature woodlands.

The results indicate that afforestation by mountain birch, and the establishment of birch woodlands, can increase the bulk SOC stock (0-30 cm), which is in accordance with the findings of Bárcena et al., (2014). They are, however, contrary to the results of Ritter (2007), who studied afforestation on heathland. Ritter (2007) published a SOC stock of about 40 t C ha$^{-1}$ (0-20 cm) for 26 and 97 year old birch stands, and thus found no change in C stocks following the succession of already vegetated heathland to birch woodland in eastern Iceland. Snorrason et al., (2002) found a higher SOC stock (0-30 cm) in a 54 year old birch stand (65 t C ha$^{-1}$), compared to that of grassland (54 t C ha$^{-1}$) at Gunnarsholt. Soil development and natural vegetation succession on moraine till after glacial retreat is another typical process of land cover change in Icleand. Vilmundardóttir et al., (2015) found a SOC accumulation within the top 20 cm from 0.9 and 13.5 t C ha$^{-1}$ at sites with a maximum age of 120 years, thereby demonstrating that the process of vegetation succession on moraine till leads to an increase in soil carbon stock. Our results indicate that the change in SOC stocks during afforestation with mountain birch on severely degraded soils (Figure 2) is comparable with those given for shrub encroachment in the cited literature.

Restoration by revegetation is another process of land cover change in Iceland (Aradóttir et al., 2000; Arnalds et al., 2013). The present study compared the effects of afforestation and revegetation. Within 50 years, the revegetated sites (Grass50), which were restored by fertilizer and grass seeds, showed a median SOC stock 9 t C ha$^{-1}$ higher compared to the soils of Birch50. This leads to the assumption that revegetation of severely degraded soils enhances the bulk SOC stock (0-30 cm) more effectively than afforestation (Figure 2). Aradóttir et al., (2000) and Snorrason et al. (2002) also studied revegetated grassland sites near Gunnarsholt showing a site history comparable to the grassland sites of the present study. Accordingly, Snorrason et al., (2002) reported a SOC stock (0-30 cm) of 54 t C ha$^{-1}$, which are comparable with the SOC stocks of the present study. However, Aradóttir et al., (2000) found 28 t C ha$^{-1}$ (0-20 cm) for a 46 year old grassland site, compared to the median value of 34 t C ha$^{-1}$ (0-20 cm) for the present grassland sites.

The SOC stocks (0-30 cm) of the land cover categories indicate that the SOC pool decreases after the establishment of birch shrubs on barren land. It then increases during tree establishment to reach the level of naturally grown birch woodlands (Figure 2). However, the analysis of SOC dynamics in such a temporally dynamic landscape, which results in unequal SOC concentrations patterns across land cover categories, calls for more detailed and alternative methods (Table 1). The present study focused on the vertical distribution of the SOC and its quality, to verify whether afforestation results in the soil becoming a C source, and whether more C is sequestered during revegetation than afforestation (see next chapter).

<< Figure 2 HERE >>



### 3.3 SOC fractionation enhances our understanding of afforestation processes

### 3.3.1 Vertical resolution of bulk SOC stocks

The vertical distribution of SOC concentrations (Table 1) and SOC stock at Grass50 with the sampled soil intervals, showed clearly that the highest SOC stock (~38 t C ha[-1]) is located between 10 and 30 cm (Figure 2). The same patterns were found

at Barren Land, which shows the unexpected but high importance of the SOC stock in deeper sampling intervals such as "10-20" and "20-30cm". Hence, two third of the calculated SOC stock was found deeper than 10 cm at Grass50. This is not in accordance with the commonly observed vertical decrease of the SOC concentration (Jobbágy and Jackson, 2000). However, this is a typical pattern of volcanic soils which are also characterized by biologicaly active soil layers buried by eratica. Arnalds and Kimble (2001) observed similar patterns for soils with lag-gravel surfaces, which developed through intense

frost heave of coarse material and aeolian deposition. Strachan et al., (1998), Snorrason et al., (2002) and Kolka-Jónsson (2011) confirm this inverse vertical SOC pattern in disturbed and undisturbed soil pedons, in the same region as the present study. Therefore, this inversion of the SOC stock with depth seems to be a common feature of sandy soils in southern Iceland, and is the result of high volcanic activity, geomorphic processes and anthropogenic disturbances (Dugmore et al., 2009; Kolka-Jónsson, 2011; Arnalds, 2015e). Hence, Andosols generally consist of chronologically layered soil horizons

with various amounts of organic carbon, as well as different densities of gravel and fine earth material which substantially influence the vertical patterns of the SOC stock.

Icelandic desert soils and severely degraded soils generally contain a SOC stock ranging from 1 to 45 t C ha[-1] before the application of any restoration activities (Óskarsson et al., 2004; Arnalds et al., 2013). The present study calculated a median SOC stock of 40 t C ha[-1] (Figure 2) for severely degraded soils, and accompanies with the higher SOC stock values given in

the cited literature. This implies that the soils of Barren Land contain a certain amount of SOC due to earlier soil formation processes prior to disturbance and SOC accumulation, and which occurred before the soil profile was cut during soil erosion processes. Nonetheless, the SOC stocks of Barren Land are significantly lower than in soils under well-established and not-degraded ecosystems (Birchnat) (Figure 2). The subdivision of the studied soil columns of 30 cm in four sampling intervals explains the higher bulk SOC stocks at Barren Land and Grass50. This is due to the higher values in the intervals "10-20

cm" and "20-30 cm" compared to the afforested birch sites (Birch15-Birch50), which constitute older buried soils (Table 1, Figure 2). The subdivision further characterizes the patterns found of bulk SOC stock (0-30 cm) (chapter 3.2), with the high bulk SOC stocks (0-30 cm) at Barren Land and Grass50 being caused by the carbon pool located deeper than 10 cm soil depth. Under the given site conditions, it is questionable to apply the commonly used soil depth of 30 cm for SOC stock monitoring (Aalde et al., 2006), to sample SOC that originates from buried soils, as it distorts the effects of restoration

activities in the results of SOC concentration and SOC stock. Based on this understanding, the bulk SOC stocks (0-30 cm) do not reveal that afforestation caused a C loss during the first 25 years of mountain birch establishment at such severely degraded sites, and that revegetation is more effective than afforestation by mountain birch within the first 50 years.





The analysis of C vertical distribution shows further that C concentration continuously increases in the top 10 cm during the establishment of birch woodland (Table 1) (Birch15-Birch50). Hence, the SOC stock increases by 10 t C ha[-1] and 3.5 t C ha[-1] in the sampling intervals "0-5 cm" and "5-10 cm" during the same time interval, respectively (Figure 2). Thus, afforestation by mountain birch on severely degraded volcanic soils is most distinct in the top 10 cm, which is comparable with the

findings of Bárcena et al. (2014). However, the SOC stock (0-10 cm) of 50-year old birch woodlands is still lower (Δ 5 t C ha[-1]) than the stocks identified at the Birchnat sites. This indicates that afforested stands can additionally accumulate SOC in the top 10 cm, during their development to mature mountain birch stands after 50 years of birch growth.

The results indicate that spatial variability must be taken into account when analyzing SOC of volcanic soils, especially when deeper than 10 cm, between the sampled sites and the land cover categories (i.e. grassland, barren, etc). Thus, the

equality or comparability of the sites, except for the studied variable, is not ensured for space-for-time substitution sampling approaches under such circumstances as performed in the present study (Walker et al., 2010). Hence, it is false to use the selected Barren Land site as initial status ($t_0$) for discussing the effect of afforestation and calculating any SOC sequestration rates. Accordingly, the authors suggest to use permanent plots and the application of long-term monitoring (Arnalds et al., 2013; Thorsson, in prep.) or cumulative coordinate approaches (Rovira et al., 2015), which seem more appropriate to assess

changes of SOC characteristics on severely degraded soils.

### 3.3.2 Analysis of soil organic carbon quality

The net primary production (NPP) of a landscape is increased during afforestation. Hence, the supply of organic material to the soil is higher at shrubby sites compared to barren areas (Aradóttir et al., 2000; Bjarnadottir et al., 2007; Arnalds et al., 2013; Hunziker et al., 2014; Vilmundardóttir et al., 2015). The mass of POM material can be taken as an indicator for this

supply. In the present study, the change of the material supply leads to an increase in the median mass of POM material (> 63 µm and < 1.8 g cm[-3]) in the top 30 cm of the soil, which was measured in the present study at Barren Land: 5; Birch15: 43; Birch20: 53; Birch25: 51; Birch50: 174 mg POM per gram soil. The sites at Birchnat contained 90 mg POM g[-1] soil. The lower value at Birchnat compared to Birch50 can be explained by the lower productivity of Birchnat due to the already undergone self-thinning process during the forest development at Birchnat. Further, the revegetation to grassland (Grass50)

showed distinctly lower median POM mass (24 mg g[-1] soil) than Birch50. According to these results, it is assumed firstly that afforestation is a more effective restoration process than revegetation with grasses, in terms of supplying organic material, and hence carbon to the soil phases and secondly, this supply increases exponentially during the establishment of afforested birch woodlands. However, this observation is inconsistent with the results of the bulk SOC stocks (0-30 cm) comparison (chapter 3.2), which suggests that the conversion of eroded land into grassland is a more effective restoration

approach. Thus, further explanations are needed to explain characteristic of these high SOC pools.

Physical fractionation of the SOC further revealed that the bulk SOC stocks at Grass50 consist mostly of carbon found in the '< 63 µm' (73 %) and HF (16 %) fractions, respectively (Table 3). Only a minor part of the SOC stock originated from the POM-fraction (3 t C ha[-1]). The vertical resolution showed further that the amount of carbon stored in the '< 63 µm' fraction



became more dominant at deeper sampling intervals at Grass 50. Hence, the bulk SOC stocks in the 10-20 and 20-30 cm layers were fed by SOC found in the '< 63 μm' fraction (Figure 2, Figure 3); results of Barren Land showed the same pattern. At these sites, the bulk SOC stock (Figure 2) consisted mostly of carbon which was stored in the '< 63 μm' (65 %) and HF (28 %) fractions, respectively (Table 3). This more detailed analysis of the SOC quality indicates that at Barren Land and Grass50 the SOC measured in deeper sampling intervals was sequestered in horizons during soil development historically. Later, these C-rich horizons of the palaeosoils were buried by aeolian transported material and then again exposed by soil erosion. This assumption of sampling material of paleaosoils is underlined by the highest allophane and ferrihydrite contents at Barren Land and Grass50 (Table 2), as a result of the weathering of soil minerals.

The combination of a vertically divided soil sampling technique and the physical SOC fractionation showed that most of the SOC at Barren Land originated in soil material which was smaller than 63 μm at a soil depth deeper than 10 cm. Hence, the SOC which is found at severely degraded soils (Óskarsson et al., 2004; Arnalds et al., 2013) seems to be 'old' buried SOC, or sedimented small-sized SOC, instead of deriving from the ongoing revegetation or succession process. This underlines the evidence that the SOC stocks measured deeper than 10 cm soil depth distort the SOC accumulation during restoration activities (previous section). Sites with such SOC patterns can therefore hardly be used as reference sites to explore the effect of restoration on SOC dynamics. The same assumption can be made for the SOC patterns at Grass50, which showed low values of POM mass and POM-C concentrations, but high C concentrations in the '< 63 μm' fraction. Thus, it is questionable whether the SOC by itself, and the difference of 17 t C ha$^{-1}$ between Barren Land and Grass50, are the result of the revegetation process. The physical fractionation revealed that the SOC found at Grass50 has rather originated from buried soil material than from revegetation. Based on this and the given results in the previous sections, it seems that afforestation is the more effective restoration process than revegetation, primarily due to the higher amount of POM material and POM-C found in the soils covered by mountain birch shrubs.

<< Table 3 HERE>>

Turning eroded land into birch woodland led to a continuous increase in the bulk SOC stock (0-30 cm) (Figure 2). During afforestation, the median stocks of the POM and '< 63 μm' fractions increased by 8 and 6 t C ha$^{-1}$ between Birch15 and Birch50, while the SOC stock of the HF fraction seemed to stagnate at about 9 t C ha$^{-1}$ (Table 3). This is explained by the increases of the '< 63 μm'-C and the POM-C concentrations during the afforested time span (Figure 3, Figure 4). During the same time span, the DOC concentration doubles. This is in accordance with Hunziker et al., (2017) who also found a doubling of the DOC concentration during the encroachment of subalpine pastures by green alder bushes.

According to the results of the present study (Figure 2, Table 3, Figure 4), afforestation by mountain birch on severely degraded soils increases the bulk SOC stock, especially in the top 10 cm. However, this increase is accompanied by a higher SOC lability, which is indicated by the increase of the POM and DOC concentrations (Figure 4), as well as the corresponding C-stocks (Table 3). Our study also found an increase of the '< 63 μm'-SOC stock of 6 t C ha$^{-1}$ between



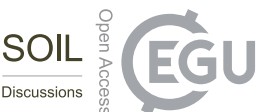

Birch15 and Birch50. This result can be attributed to a stabilization of the SOC due to its binding with clay minerals in this fraction, which contains material of silt and clay size. However, the extraction of the material of the '< 63 μm' fraction by the physical separation technique of Zimmermann et al., (2007), and the measurement of its C concentration, hardly provide an indication for the stability of the SOC in the analyzed fraction due to the separation by only wet-sieving. Hence, the

method does not give any information about the location of the organic matter in the '< 63 μm' fraction and consequently, the degree of the SOC stabilization. Since the formation of organo-mineral complexes is pH dependent, the study does not give evidence whether i) the majority of the SOC is bounded to the mineral phase of the '< 63 μm' fraction, or ii) in the '< 63 μm' fraction, where the majority of the SOC is POM material that is disconnected from the mineral phases.

<< Figure 3 HERE >>

<< Figure 4 HERE >>

### 3.4 SOC stabilization by volcanic clay minerals

Clay minerals found in volcanic soils, such as those found in Iceland, may play a key role in stabilizing soil organic carbon due to their amorphism, high degree of hydration, extensive specific surface area (200-1500 $m^2$ $g^{-1}$), and pH-dependent charge and the high reactivity (Torn et al., 1997; Basile-Doelsch et al., 2007; McDaniel et al., 2012; Arnalds, 2015a). The major stabilization mechanisms are either the formation of allophane- or ferrihydrite- humus complexes, which is favored at pH > 5.0, or the building of metal-humus complexes which are more effective at pH values lower than 5.0 (Arnalds, 2015a).

The $Al_{pyr}:Al_{ox}$ and $Fe_{pyr}:Fe_{ox}$ ratios (Table 1) are used as an indicator of the occurrence of metal-humus complexes. The higher the ratio, the more clay minerals are bound to organic compounds, which suggests an increase of the SOC stabilization. In order to discuss stabilization processes of SOC found in the '< 63 μm' fraction' with mineral clays (chapter 3.3), we therefore considered the SOC concentration of the '< 63 μm' fraction in Figure 5.

In general, the results showed a decline of the $Al_{pyr}:Al_{ox}$ and $Fe_{pyr}:Al_{ox}$ ratios with soil depth for all land cover categories

(Table 2). Further, Birchnat and Birch50 showed the highest $Al_{pyr}:Al_{ox}$ and $Fe_{pyr}:Fe_{ox}$ ratios, while at Barren Land and Grass50, the lowest ratios were found (Table 2). The present study found positive ($r^2 = 0.44$) and negative ($r^2 = -0.60$) correlations between the allophane concentration and the $(Al_{pyr}+Fe_{pyr}):(Al_{ox}+Fe_{ox})$ ratio and pH value, respectively (Figure 5; A, D). The allophane concentrations are highest in the soils which were un-vegetated (Barren Land). On the other hand, the concentrations of Al and Fe bound to metal-humus complexes were highest in the top sampling intervals of sites with the

longest vegetation covers (Birchnat, Birch50 and Birch25; dotted circle). Both correlations indicate a possible influence of the different stages of vegetation cover on the amounts of allophane and the ratio $Al_{pyr}:Fe_{pyr}$, respectively. This can be





explained by the increase in protons resulting from vegetation processes in the soil, which leads to acidification and simultaneously a lowering of the pH. Hence, the establishment of vegetation favors the formation of metal-humus complexes (Arnalds, 2008, 2015c).

The scatterplots comparing the allophane concentrations with the bulk SOC concentrations as well as the '< 63 μm' SOC

concentrations, show no clear trends as most of the samples contained < 4 % of bulk SOC or '< 63 μm' SOC. The highest SOC concentrations were found in the upper sampling intervals at Birch25, Birch50 and Birchnat (dotted circles). However, the allophane content is lowest in these cases (dotted circle), which may be attributed to the fact that soil weathering and the formation of clay minerals takes longer than the allocation of soil organic carbon during birch growth. Regarding SOC sequestration during the reclamation of severely degraded land and soils, soil material of eroded and capped soil profiles

most likely already passed through weathering processes, and therefore contained a high amount of clay minerals. Hence, the carbon sequestration potential of these eroded soils may be relatively high (Arnalds et al., 2000; Ágústsdóttir, 2004). The fresh SOC originating from reclamation activities can be stabilized by the already existing clay minerals like allophane (Table 2, Figure 5). The stabilization of the SOC in the form of metal-humus complexes seems to be undetermined (Figure 5; E, F). As already mentioned, the upper most sampling intervals of the vegetated sites (dotted circle) are decoupled from the

nested scatters (Figure 5; E, F). For the soil intervals of these sites (circled), the formation of metal-humus complexes might comprise a reasonable stabilization process of the SOC in the '< 63 μm' fraction due to its positive relationship (Figure 5; F). The analysis of the relation between the SOC and volcanic minerals indicates that during afforestation, the organic carbon is preferably stabilized in metal-humus complexes. It implies that this process starts to be an effective stabilization process for bulk SOC and '< 63 μm'-SOC after 20 years of birch growth, and can be found in deeper sampling intervals in older birch

stands.

<< Figure 5 HERE >>

## 5 Conclusions

The study aimed to evaluate the SOC sequestration potential of afforestation on severely degraded soils in southern Iceland. For this, we measured the SOC stocks of differently-aged afforested birch stands and compared them with those of eroded and degraded soils, re-vegetated grasslands and non-degraded woodlands which have escaped the soil erosion, respectively. In addition, the SOC quality of all sites was analyzed by physical soil fractionation.

Afforestation with mountain birch leads to a continuous increase of the SOC stock (15 t C ha$^{-1}$) for birch stands between 15

and 50 years. Afforested birch stands can still potentially accumulate SOC after 50 years of growth, due to their lower SOC stock ($\Delta$ = 13 t C ha$^{-1}$) compared to naturally, old growth birch woodlands. The POM mass and POM-C concentrations increase during the succession of the mountain birch ecosystem. However, due to the increased amount of POM-C stock and

the doubling of the DOC stock, it seems that afforestation leads to bulk SOC pools which are more vulnerable to release C to the atmosphere.

The results of the present study also clearly show that undertaking research on soil organic carbon patterns on severely degraded soils within this area is challenging, owing to the high bulk SOC stocks (0-30 cm) of these degraded soils. The

present study differentiated between the physically separated SOC pools, which allowed for the evaluation of the success of afforestation by mountain birch on a landscape with highly diverse soil patterns and SOC distributions. Considerable amounts of the bulk SOC stocks can be found between 10 and 30 cm, even in severely degraded soils, while most of the SOC is found in the '< 63 µm' fraction. These findings indicate that the soils already contain certain amounts SOC which is not related to any vegetation restoration process, but instead to old buried soil horizons. Hence, the applied space-for-time

substitution approach showed limited success by reason of the heterogeneity of the parent material, and its SOC properties at greater soil depths. In such cases, it would be more effective to use permanent plots and a long-term monitoring approaches to assess soil development during vegetation restoration, as initially suggested by Johnson and Miyanishi (2008), carried out by Arnalds et al., (2013) and Thorsson (in prep.), and further developed by Bárcena et al., (2014). A key message of the study is that the standardized soil sampling depth of 30 cm turns out to be questionable for evaluating the success of

restoration regarding SOC sequestration on severely degraded soils, thus, it needs to be applied with caution.

**Author contribution**

Matthias Hunziker designed the sampling setup, did the soil sampling and led the lab analysis procedure. Matthias Hunziker also did the statistics and prepared the manuscript with valuable contributions of the two co-authors Olafur Arnalds and Nikolaus J. Kuhn. Nikolaus J. Kuhn provided the lab facilities and supervised Matthias Hunziker during his PhD studies.

Olafur Arnalds was the project leader of the research project CarbBirch.

**Acknowledgements**

This work contributes to the CarbBirch project funded by Orkuveita Reykjavikur and the work within the Nordic Centre of Advanced Research on Environmental Services (CAR-ES) and the Forest Soil C-sink Nordic Network (FSC-Sink). We want to thank our lab technician and friend Marianne Caroni, who sadly left us much too early for her help and inspired

discussions. We would also like to extend our gratitude to Ruth Strunk and Judith Kobler for their help in the laboratory during carbon and volcanic clay measurements. Nina Carle and Mathias Würsch helped during gathering data in the field and in the laboratory. Our sincerest thank goes to Gudmundur Halldorsson and the people of the Soil Convervation Service at Gunnersholt for their help and hospitality. Further, the authors gratefully acknowledge Vladimir Wingate for improving the grammar.



**Competing interests**

The authors declare that they have no conflict of interest.




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

| Type | Depth | Volume gravel (> 2 mm) | Bulk density (< 2 mm) | C content | C:N ratio | pH (H$_2$O) | pH (KCl) |
|------|-------|------------------------|------------------------|-----------|-----------|-------------|----------|
| | [cm] | [cm$^3$ 100 cm$^{-3}$] | [g cm$^{-3}$] | [%] | [-] | [-] | [-] |
| Barren Land | 0-5 | 11.8 (1.3; 14.5) | 0.76 (0.64; 0.82) | 1.7 (0.9; 2.9) | 10.7 (9.9; 13.5) | 7.0 (6.7; 7.1) | 5.7 (5.4; 5.7) |
| | 5-10 | 5.0 (0.5; 11.3) | 0.65 (0.60; 0.82) | 3.1 (0.9; 3.2) | 13.3 (10.1; 14.8) | 7.2 (7.0; 7.2) | 5.8 (5.6; 5.8) |
| | 10-20 | 0.5 (0.3; 9.3) | 0.54 (0.49; 0.79) | 1.7 (1.1; 2.4) | 12.1 (10.6; 13.3) | 7.2 (7.0; 7.2) | 5.8 (5.7; 5.9) |
| | 20-30 | 2.8 (0.5; 3.8) | 0.48 (0.48; 0.56) | 2.7 (2.2; 2.8) | 13.5 (11.1; 14.5) | 7.3 (6.8; 7.3) | 5.9 (5.5; 5.9) |
| Birch15 | 0-5 | 6.8 (1.0; 8.3) | 0.75 (0.66; 0.85) | 2.1 (1.4; 2.4) | 15.2 (14.4; 17.5) | 6.1 (6.0; 6.4) | 4.9 (4.8; 5.0) |
| | 5-10 | 4.5 (1.3; 5.3) | 0.87 (0.80; 0.89) | 0.9 (0.9; 1.3) | 11.5 (11.5; 12.7) | 6.6 (6.3; 6.6) | 5.2 (5.0; 5.2) |
| | 10-20 | 3.0 (1.0; 4.8) | 0.89 (0.69; 0.91) | 1.1 (0.6; 2.0) | 10.7 (10.5; 12.1) | 6.8 (6.7; 6.8) | 5.3 (5.2; 5.4) |
| | 20-30 | 4.0 (0.0; 8.3) | 0.76 (0.56; 0.90) | 1.1 (0.4; 2.8) | 11.1 (10.0; 11.9) | 6.9 (6.8; 6.9) | 5.5 (5.4; 5.5) |
| Birch20 | 0-5 | 1.5 (0.8; 4.3) | 0.55 (0.47; 0.69) | 2.9 (2.1; 5.0) | 15.6 (15.6; 17.2) | 6.1 (6.0; 6.2) | 4.9 (4.9; 5.0) |
| | 5-10 | 1.0 (0.5; 3.0) | 0.79 (0.66; 0.89) | 1.5 (0.8; 2.0) | 12.0 (10.1; 13.7) | 6.5 (6.4; 6.6) | 5.1 (5.1; 5.3) |
| | 10-20 | 1.0 (0.5; 3.3) | 0.82 (0.69; 0.89) | 1.1 (0.7; 1.7) | 11.1 (10.1; 12.8) | 6.7 (6.6; 6.9) | 5.3 (5.2; 5.5) |
| | 20-30 | 1.3 (0.3; 5.0) | 0.91 (0.66; 0.95) | 1.1 (0.8; 1.8) | 11.2 (10.6; 14.7) | 6.8 (6.8; 7.0) | 5.3 (5.3; 5.6) |
| Birch25 | 0-5 | 2.3 (1.0; 8.0) | 0.59 (0.44; 0.76) | 3.4 (2.1; 5.5) | 15.9 (14.1; 17.0) | 6.1 (6.0; 6.3) | 5.0 (5.0; 5.1) |
| | 5-10 | 0.5 (0.4; 2.5) | 0.77 (0.75; 0.89) | 1.8 (1.0; 2.0) | 12.2 (11.5; 13.3) | 6.5 (6.5; 6.7) | 5.2 (5.2; 5.2) |
| | 10-20 | 3.0 (0.3; 3.0) | 0.82 (0.80; 0.89) | 1.1 (1.0; 1.5) | 11.1 (10.9; 11.9) | 6.7 (6.7; 6.7) | 5.3 (5.2; 5.4) |
| | 20-30 | 0.5 (0.1; 1.8) | 0.79 (0.74; 0.90) | 1.4 (1.0; 1.7) | 11.0 (10.4; 11.3) | 6.7 (6.7; 6.8) | 5.3 (5.3; 5.4) |





6    **Continued**

| Type | Depth | Volume gravel (> 2 mm) | Bulk density (< 2 mm) | C content | C:N ratio | pH (H$_2$O) | pH (KCl) |
|---|---|---|---|---|---|---|---|
| | [cm] | [cm$^3$ 100 cm$^{-3}$] | [g cm$^{-3}$] | [%] | [-] | [-] | [-] |
| Birch50 | 0-5 | 1.0 (1.0; 1.8) | 0.44 (0.40; 0.49) | 8.1 (5.5; 9.8) | 18.6 (16.9; 20.9) | 5.8 (5.8; 6.0) | 4.8 (4.8; 4.8) |
| | 5-10 | 0.8 (0.5; 2.5) | 0.75 (0.68; 0.78) | 1.9 (1.7; 2.4) | 12.7 (12.5; 13.7) | 6.3 (6.3; 6.4) | 5.0 (5.0; 5.0) |
| | 10-20 | 3.0 (1.3; 4.5) | 0.78 (0.72; 0.84) | 1.5 (1.1; 1.8) | 11.8 (10.9; 12.2) | 6.5 (6.1; 6.5) | 5.1 (5.1; 5.2) |
| | 20-30 | 0.3 (0.1; 1.1) | 0.88 (0.85; 0.90) | 1.1 (0.9; 1.3) | 10.7 (10.6; 11.9) | 6.6 (6.6; 6.9) | 5.2 (5.1; 5.3) |
| Grass50 | 0-5 | 6.3 (6.3; 8.8) | 0.72 (0.68; 0.73) | 2.5 (2.5; 2.8) | 12.6 (12.5; 13.0) | 6.4 (6.3; 6.5) | 5.1 (5.0; 5.1) |
| | 5-10 | 7.5 (5.0; 7.5) | 0.85 (0.71; 0.86) | 2.0 (1.2; 2.3) | 11.2 (10.7; 11.4) | 6.7 (6.7; 6.7) | 5.4 (5.2; 5.4) |
| | 10-20 | 5.5 (3.8; 13.8) | 0.63 (0.63; 0.75) | 2.6 (1.8; 3.4) | 10.9 (10.5; 11.4) | 6.8 (6.8; 6.9) | 5.5 (5.5; 5.5) |
| | 20-30 | 1.5 (0.2; 7.5) | 0.63 (0.61; 0.76) | 3.4 (2.0; 3.4) | 12.2 (12.1; 12.5) | 7.0 (6.9; 7.1) | 5.6 (5.5; 5.8) |
| Birchnat | 0-5 | 0.5 (0.5; 0.5) | 0.51 (0.46; 0.52) | 6.3 (6.3; 6.5) | 19.2 (19.0; 19.2) | 6.0 (6.0; 6.2) | 5.0 (5.0; 5.1) |
| | 5-10 | 0.3 (0.1; 0.6) | 0.68 (0.64; 0.68) | 4.0 (3.3; 5.1) | 16.3 (16.1; 17.7) | 6.3 (6.3; 6.4) | 5.1 (5.0; 5.1) |
| | 10-20 | 0.1 (0.0; 0.1) | 0.67 (0.64; 0.67) | 2.4 (2.1; 3.4) | 13.7 (13.2; 15.7) | 6.5 (6.2; 6.6) | 5.2 (5.0; 5.3) |
| | 20-30 | 0.1 (0.0; 0.3) | 0.72 (0.70; 0.76) | 1.9 (1.8; 2.0) | 12.6 (11.8; 12.8) | 6.7 (6.7; 6.8) | 5.3 (5.2; 5.4) |

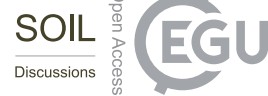



Table 2: Volcanic soil properties of the studied vegetation types and sampled depth intervals. The median value and the minimum and maximum values (in paranthesis) are given as above.

| Type | Depth | $Al_{pyr}$ | $Al_{pyr} : Al_{ox}$ ratio | $Fe_{pyr}$ | $Fe_{pyr} : Fe_{ox}$ ratio | $(Al + \frac{1}{2}Fe)_{ox}$ | Allophane | Allophane + Ferrhydrite Clay |
|---|---|---|---|---|---|---|---|---|
| | [cm] | [%] | [$10^1$, -] | [%] | [$10^1$, -] | [%] | [%] | [%] |
| Barren | 0-5 | 0.20 (0.14; 0.27) | 0.82 (0.72; 1.07) | 0.17 (0.13; 0.23) | 0.42 (0.39; 0.48) | 4.50 (2.65; 6.65) | 13.9 (8.2; 20.2) | 20.8 (12.7; 30.3) |
| Land | 5-10 | 0.26 (0.13; 0.27) | 0.63 (0.60; 0.90) | 0.22 (0.13; 0.24) | 0.36 (0.34; 0.47) | 7.46 (2.81; 7.71) | 22.7 (9.3; 23.7) | 33.9 (13.9; 34.8) |
| | 10-20 | 0.16 (0.14; 0.17) | 0.79 (0.54; 0.85) | 0.18 (0.14; 0.19) | 0.47 (0.35; 0.50) | 3.90 (3.17; 5.85) | 12.4 (10.4; 18.7) | 18.6 (15.6; 27.8) |
| | 20-30 | 0.24 (0.15; 0.28) | 0.57 (0.44; 1.01) | 0.23 (0.21; 0.29) | 0.37 (0.35; 0.65) | 6.51 (5.03; 7.31) | 21.7 (15.2; 23.2) | 31.8 (22.9; 33.8) |
| Birch15 | 0-5 | 0.26 (0.20; 0.26) | 1.76 (1.55; 1.90) | 0.21 (0.15; 0.21) | 0.69 (0.55; 0.71) | 2.82 (2.60; 3.01) | 8.6 (8.2; 9.5) | 13.6 (12.7; 14.6) |
| | 5-10 | 0.17 (0.17; 0.19) | 1.23 (1.13; 1.24) | 0.14 (0.13; 0.15) | 0.47 (0.41; 0.50) | 2.96 (2.95; 3.05) | 9.5 (9.4; 9.8) | 14.8 (14.6; 14.8) |
| | 10-20 | 0.19 (0.13; 0.27) | 1.00 (0.95; 1.11) | 0.17 (0.11; 0.25) | 0.49 (0.36; 0.50) | 3.37 (2.73; 5.35) | 10.8 (9.5; 17.0) | 16.5 (14.5; 25.7) |
| | 20-30 | 0.18 (0.09; 0.31) | 0.8 (0.75; 1.08) | 0.15 (0.09; 0.36) | 0.49 (0.34; 0.52) | 3.26 (2.51; 7.59) | 10.5 (8.8; 24.7) | 15.9 (13.5; 36.5) |
| Birch20 | 0-5 | 0.33 (0.26; 0.36) | 1.67 (1.44; 2.21) | 0.28 (0.24; 0.41) | 0.77 (0.64; 1.37) | 3.15 (3.13; 4.49) | 9.5 (8.9; 13.4) | 14.9 (14.0; 20.8) |
| | 5-10 | 0.21 (0.16; 0.26) | 1.09 (0.95; 1.19) | 0.18 (0.13; 0.25) | 0.51 (0.44; 0.53) | 3.52 (2.93; 5.09) | 11.4 (9.5; 16.3) | 17.3 (14.6; 24.4) |
| | 10-20 | 0.17 (0.16; 0.20) | 0.97 (0.85; 1.00) | 0.16 (0.15; 0.21) | 0.47 (0.46; 0.52) | 3.56 (3.16; 4.44) | 12.0 (11.1; 14.4) | 17.9 (16.5; 21.5) |
| | 20-30 | 0.19 (0.14; 0.19) | 0.92 (0.69; 0.96) | 0.19 (0.15; 0.23) | 0.48 (0.47; 0.49) | 3.99 (3.14; 5.11) | 13.6 (11.0; 17.5) | 20.3 (16.4; 25.6) |
| Birch25 | 0-5 | 0.32 (0.26; 0.38) | 1.70 (1.64; 2.23) | 0.32 (0.24; 0.42) | 0.91 (0.73; 1.26) | 3.37 (3.18; 3.65) | 10.1 (10.0; 10.9) | 15.6 (15.6; 17.0) |
| | 5-10 | 0.23 (0.18; 0.25) | 1.22 (1.09; 1.23) | 0.20 (0.16; 0.22) | 0.55 (0.48; 0.57) | 3.76 (3.34; 3.94) | 11.9 (11.3; 12.7) | 18.1 (17.0; 19.2) |
| | 10-20 | 0.18 (0.17; 0.22) | 1.01 (0.92; 1.1) | 0.16 (0.15; 0.19) | 0.47 (0.41; 0.50) | 3.81 (3.36; 3.98) | 12.7 (11.6; 12.9) | 19.0 (17.4; 19.5) |
| | 20-30 | 0.22 (0.16; 0.24) | 1.12 (0.89; 1.12) | 0.20 (0.15; 0.22) | 0.52 (0.42; 0.53) | 3.92 (3.52; 4.23) | 12.8 (11.8; 14.0) | 19.3 (17.8; 21.1) |





13  **Continued**

| Type | Depth | $Al_{pyr}$ | $Al_{pyr} : Al_{ox}$ ratio | $Fe_{pyr}$ | $Fe_{pyr} : Fe_{ox}$ ratio | $(Al + \frac{1}{2}Fe)_{ox}$ | Allophane | Allophane + Ferrhydrite Clay |
|---|---|---|---|---|---|---|---|---|
| | [cm] | [%] | $[10^1, -]$ | [%] | $[10^1, -]$ | [%] | [%] | [%] |
| Birch50 | 0-5 | 0.52 (0.45; 0.58) | 2.83 (2.81; 3.12) | 0.65 (0.58; 0.72) | 1.92 (1.86; 2.01) | 3.57 (3.13; 3.65) | 9.2 (8.4; 10.1) | 15.1 (13.6; 16.2) |
| | 5-10 | 0.30 (0.26; 0.34) | 1.54 (1.26; 1.72) | 0.28 (0.22; 0.31) | 0.76 (0.55; 0.82) | 3.89 (3.79; 4.08) | 11.7 (11.5; 12.6) | 18.0 (17.9; 19.4) |
| | 10-20 | 0.24 (0.20; 0.25) | 1.13 (1.13; 1.14) | 0.21 (0.18; 0.22) | 0.53 (0.50; 0.58) | 4.08 (3.44; 4.20) | 12.4 (11.5; 13.5) | 18.6 (17.3; 20.5) |
| | 20-30 | 0.19 (0.14; 0.19) | 1.00 (0.86; 1.02) | 0.18 (0.14; 0.18) | 0.48 (0.43; 0.54) | 3.55 (3.25; 3.74) | 11.2 (11.0; 12.3) | 16.8 (16.5; 18.6) |
| Grass50 | 0-5 | 0.29 (0.28; 0.29) | 1.44 (1.33; 1.65) | 0.24 (0.24; 0.26) | 0.71 (0.66; 0.80) | 3.62 (3.38; 4.03) | 10.4 (10.0; 11.9) | 16.2 (15.5; 18.1) |
| | 5-10 | 0.24 (0.19; 0.25) | 1.12 (1.09; 1.15) | 0.20 (0.18; 0.25) | 0.55 (0.53; 0.62) | 4.10 (3.27; 4.30) | 12.0 (10.3; 12.9) | 18.5 (15.7; 19.6) |
| | 10-20 | 0.31 (0.25; 0.34) | 1.23 (1.23; 1.29) | 0.25 (0.24; 0.32) | 0.65 (0.58; 0.73) | 4.67 (3.93; 4.82) | 13.1 (12.0; 13.5) | 20.3 (18.3; 21.1) |
| | 20-30 | 0.26 (0.26; 0.32) | 1.03 (1.00; 1.09) | 0.30 (0.26; 0.33) | 0.61 (0.58; 0.74) | 4.77 (4.75; 5.41) | 15.6 (13.5; 15.9) | 23.1 (21.0; 24.3) |
| Birchnat | 0-5 | 0.33 (0.30; 0.37) | 2.81 (2.54; 2.93) | 0.44 (0.42; 0.49) | 2.18 (1.93; 2.21) | 2.27 (2.14; 2.45) | 5.9 (5.6; 6.5) | 9.6 (9.0; 10.3) |
| | 5-10 | 0.33 (0.32; 0.39) | 2.34 (2.15; 2.66) | 0.40 (0.39; 0.52) | 1.68 (1.47; 2.05) | 2.76 (2.52; 2.87) | 7.4 (6.9; 7.9) | 11.8 (10.9; 12.5) |
| | 10-20 | 0.27 (0.24; 0.29) | 1.49 (1.27; 1.51) | 0.28 (0.25; 0.32) | 0.88 (0.79; 1.06) | 3.47 (3.33; 3.54) | 10.2 (9.4; 10.3) | 15.6 (14.6; 15.7) |
| | 20-30 | 0.25 (0.24; 0.27) | 1.20 (1.20; 1.34) | 0.25 (0.24; 0.27) | 0.74 (0.70; 0.83) | 3.67 (3.63; 3.81) | 10.9 (10.8; 11.5) | 16.6 (16.4; 17.3) |





**Table 3: The SOC stocks [t C ha$^{-1}$] at the 0-30 cm layer, explained by SOC fractions. The median value and the minimum and maximum values (in paranthesis) are given.**

| Type | SOC stock | | | |
|------|-----------|---|---|---|
| | POM | HF | < 63 µm | DOC |
| | [t C ha$^{-1}$] | [t C ha$^{-1}$] | [t C ha$^{-1}$] | [$10^1$ t C ha$^{-1}$] |
| Barren Land | 0.7 (0.2; 1.3) | 11.5 (9.1; 13.0) | 26.4 (14.2; 33.3) | 2.0 (1.2; 2.7) |
| Birch15 | 4.9 (3.5; 7.6) | 7.3 (5.5; 11.9) | 17.4 (5.9; 25.3) | 1.1 (0.6; 1.9) |
| Birch20 | 5.8 (3.6; 9.2) | 9.9 (7.6; 11.9) | 16.5 (12.3; 20.9) | 1.1 (0.8; 2.1) |
| Birch25 | 6.3 (3.2; 9.3) | 8.7 (7.1; 9.8) | 19.9 (16.3; 26.0) | 1.4 (1.1; 2.1) |
| Birch50 | 13.2 (7.1; 17.3) | 9.0 (8.2; 9.8) | 23.5 (18.3; 25.5) | 2.5 (1.7; 3.1) |
| Grass50 | 3.1 (1.7; 4.0) | 9.1 (8.0; 10.5) | 41.5 (25.8; 53.9) | 2.8 (1.7; 3.3) |
| Birchnat | 11.5 (9.5; 16.2) | 12.8 (10.3; 19.8) | 32.4 (29.5; 36.8) | 3.7 (3.1; 5.3) |



Figures and figure captions



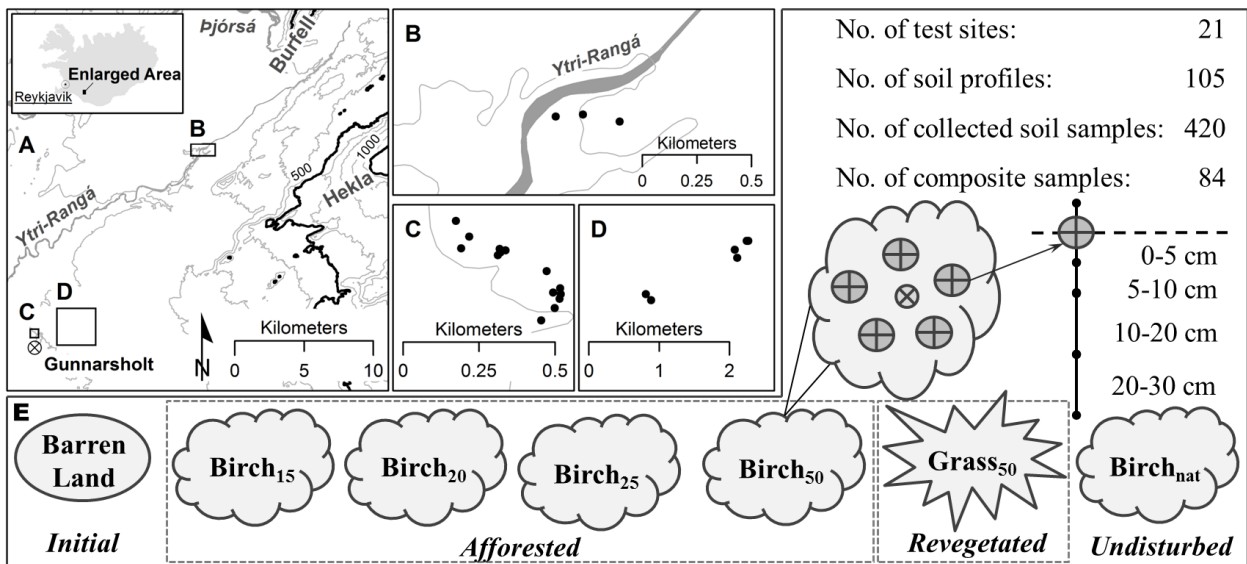

**Figure 1: The topological map (equidistance = 100 m) showing the study area and Gunnarsholt (crossed cycle) in the south of Iceland (A). The locations of the naturally growing birch woodland (B; asterisks) and the afforested (C; B15: circles, B20: triangles; B25: pentagons; B50: diamonds) and degraded (crosses) as well the revegetated (stars) test sites (D) are shown in more detail. The sampling scheme illustrates the age and vegetation characteristics of the different study sites and the applied soil sampling setup (E).**





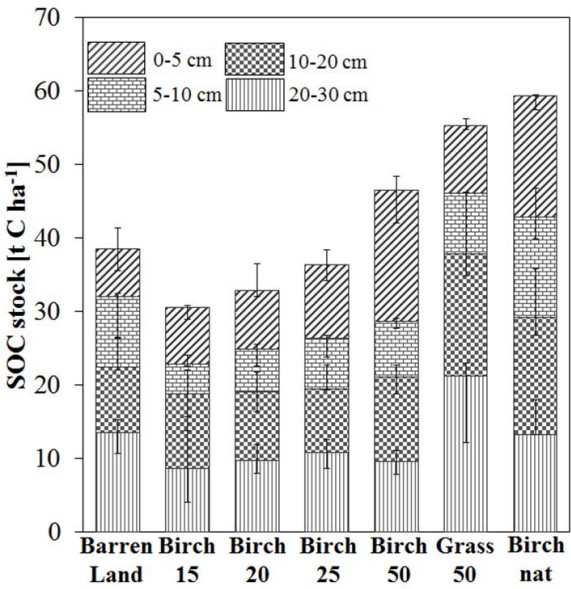

**Figure 2: Median soil organic carbon stocks [t C ha⁻¹] in the mineral soil of the studied eroded (Barren Land), reclaimed (Grass50, Birch15, Birch20, Birch25 and Birch50) and old-growth (Birchnat) sites. The range of the error bars is shows the minimum and maximum values. The different shadings indicate the four sampling depths (0-5cm: diagonal lines; 5-10cm: rectangular squares; 10-20cm: b,w squares; 20-30cm: vertical lines).**





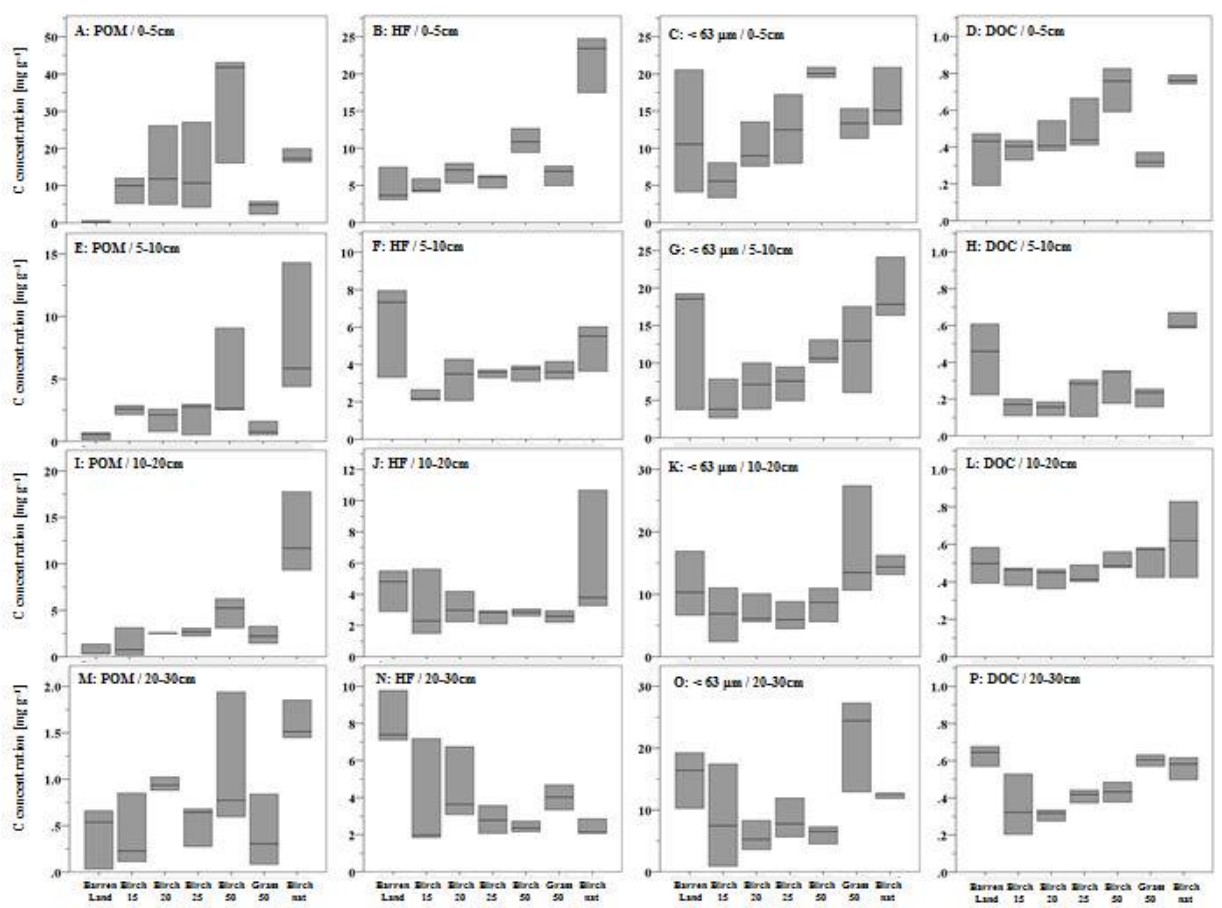

**Figure 3: SOC concentration [mg g⁻¹] of the fraction POM (A, E, I, M), HF (B, F, J, N), '< 63 µm' (C, G, K, O) and DOC (D, H, L, P) divided into the sampled soil depths (0-5, 5-10, 10-20 and 20-30 cm) for the reclaimed (Birch15, Birch20, Birch25, Birch50 and Grass50), eroded (Barren Land) and old-growth (Birchnat) sites. The boxes are show the minimum, median and maximum values. Note the variable scale of the Y-axis.**

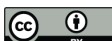



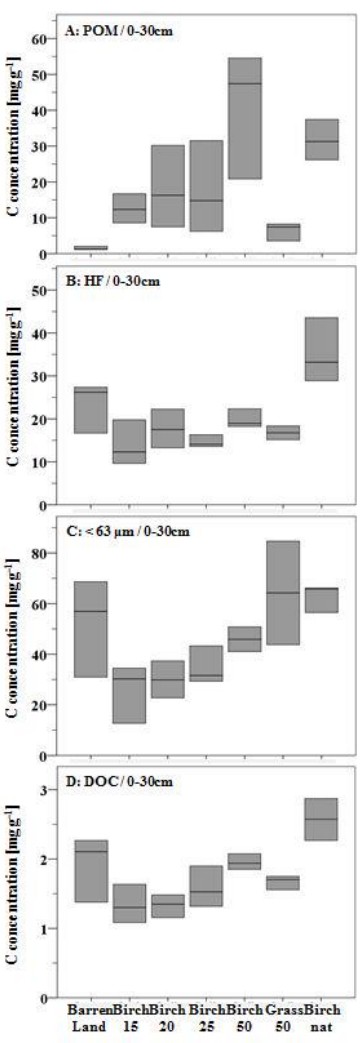

**Figure 4: Cumulated carbon concentrations [mg g⁻¹] (0-30cm) within the analyzed SOC fractions for the reclaimed (Birch15, Birch20, Birch25, Birch50 and Grass50), eroded (Barren Land) and old-growth (Birchnat) sites. The boxes show the minimum, median and maximum values. Note the variable scale of the Y-axis.**

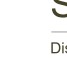



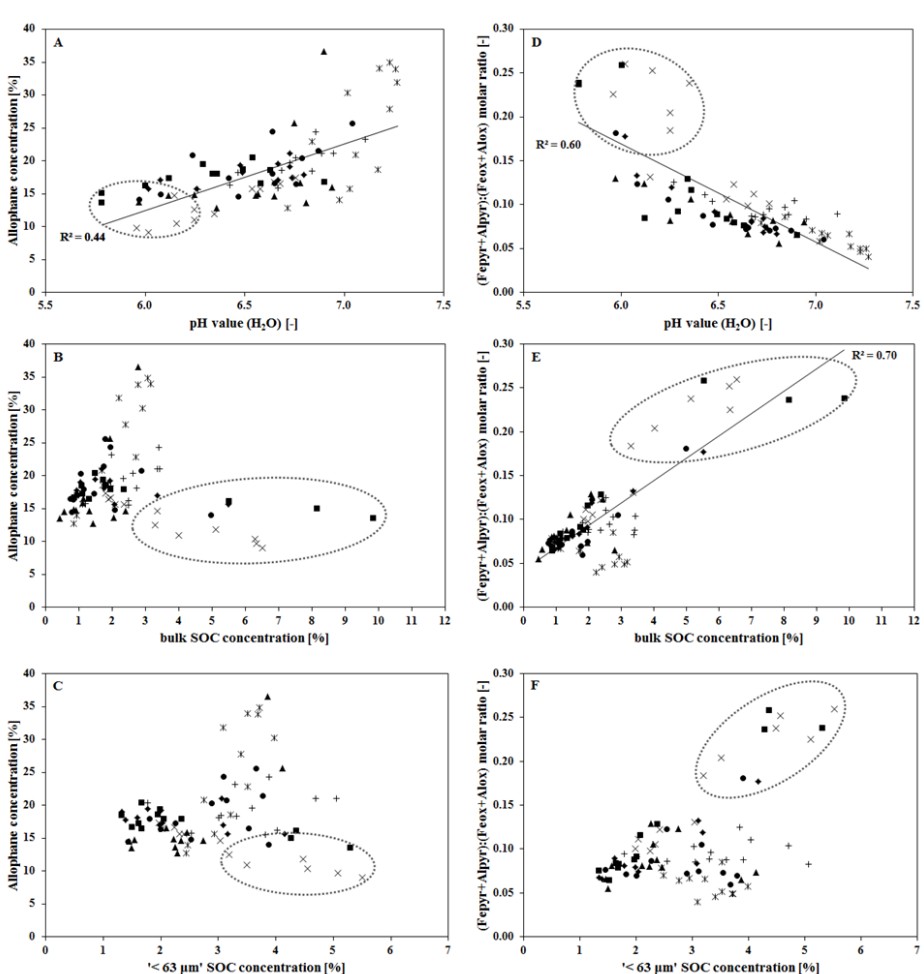

**Figure 5: Relationship between common properties of volcanic soils. The charts show the allophane concentration [%] as a function of pH value (H₂O) [-] (A), bulk SOC concentration [%] (B) and '< 63 µm' SOC concentration [%] (C), as well as the amount of Al and Fe, in the form of organo-mineral complexes ((Fe$_{pyr}$+Al$_{pyr}$):(Fe$_{ox}$+Al$_{ox}$) molar ratio [-]), as a function of pH value (H₂O) [-] (D), bulk SOC concentration [%] (E) and '< 63 µm' SOC concentration [%] (F). The observations (N = 84) are labelled based on the vegetation types: Barren Land (✳), Birch15 (▲), Birch20 (●), Birch25 (◆), Birch50 (■), Grass50 (+) and Birchnat (×). The dotted circles show all samples of Birchnat (0-5 cm, 5-10 cm), all samples of Birch50 (0-5 cm) and one sample of Birch25 (0-5 cm).**