# Peer review of "Evaluating the carbon sequestration potential of volcanic soils in South Iceland after birch afforestation"

_SOIL, 2018_

## Referee Comment (RC1) · L. Menichetti (Referee) · 15 Aug 2018

**General comment**

The study is conducted in one of the most fascinating setups ever (and this means something to me, I have to admit my bias here. I am pretty interested in these data also because of a fascination for Iceland on one side and for Andosols on another side). It also contains some pretty interesting data, and their impact can be substantial. There is a lot of work behind the data, and there are a lot of points and really long-term treatments (also extremely interesting treatments by the way! I would personally want to know more about the Barren Land treatment, but the experimental design seems in

general fantastic, with really long treatments although with the problem of underlying soil variability). There seems to be a lot of valuable information here. The manuscript nevertheless falls shortly in making the most out of such data. It lacks statistical tests, and hypotheses are not discussed in a testing framework. You could and should work more on it, in my opinion.

You should restructure a bit your hypotheses and conclusions. You mention in the text a bit too many times the same things, that SOC below 10 cm was surprising, it might be due to soil characteristic and it was due (you suppose) to previous accumulation and fossil soil layers. It all makes sense to me, and I understand this was unexpected for you, but try to make some logical blocks for it.

After you cleared the hypothesis testing framework, then you can work on the statistics for comparison, and clarify each comparison. Compare something to something else, always. If something increases, always states compared to what it increases. And put there the results from a statistical test for the comparison. You do not need to use necessarily fancy tests, just the basics t-test, ANOVA, linear regressions, combined in different ways, could be enough as tools. But use them extensively and ideally do not state anything without being able to offer some statistical evidence (there are of course exceptions, but in your case they look more like the rule).

As a suggestion for future developments, I think you should think to some more detailed modelling, at least with some compartmental model with analytically solved steady state, and try to model the input functions (I attached a reference that might be useful as a rough start). If you get right the function of input variation, a SOC model can be calibrated on your data and give you the change of the steady state. . . and predict the future steady state, meaning the C you could accumulate in these soils. But this is clearly outside the scope of your present study, I just got excited about the idea. . . and to me such idea is exemplifying why I do believe you have a lot of useful information here.

**Specific comments**

Abstract: The idea of C "sequestered as labile" sounds a bit counterintuitive in itself to me... I'm not sure I would consider labile C as "sequestered", since it stays for very short time anyway. But this is philosophy after all, not a major comment.

Line 11, page 3: Define "sequestration rate" better. The inputs in an afforestation follow a function variable over time, and this causes a continuous variation in the rate of change of C stored. I am sure this is what you meant, but "sequestration rate" was not defined before and it might be ambiguous. Is the "sequestration factor", right? Yes, that would be variable over time following the function of input variation.

Line 2, page 6: you do not complete the Zimmermann fractionation! You skip the last step, the oxidation, right? You need to mention this in M&M. This is pretty crucial to understand many of your following statements (I initially missed this detail and I read almost 2/3rd of the manuscript before realizing it).

Line 5-6, page 7: I guess you mean "each soil fraction", giving an example for the fine soil fraction only, right? But the equation was the same for all fractions, right?

Line 28, page 7: this is confusing. You sampled also "Barren Land", no? Whichhad a high ferrhydrite content, as you say before. High compared to what? Do you mean that ferrhydrite was "high", but allophane material was higher than this latter? Or do you measn "all other samples than Barren Land"?

Line 1, page 8: what do you mean with "nutrient contents"? Other nutrients than C? Please specify which nutrients. And in Table 1 you report only the C:N but not N, it is quite difficult to read this statement in the data (I would need to extract the values and do the calculation from C and C:N ratio). If also N was important, please report it with direct measurements, otherwise talk only about C.

Line 13, page 8: maybe you cannot generalize so much. Only the degraded volcanic soil soils you sampled showed that, I would say you cannot say the same in general.

line 26, page 8: "usually tested" is generic term that has pretty much no meaning. By whom? Maybe my usual tests are not the same, maybe I am personally used to something else?

Line 3-13, page 9: all this discussion has the defect of not considering the starting point. Soil C stocks are an equilibrium defined over several decades/centuries by the inputs. A land use change represents a change from a possible equilibrium state to something else, in theory a new equilibrium after climax. If this means a loss or a gain of SOC depends on the new inputs (so the age of the plantation in this case), but also on what was there before. In all these comparisons you should indicate at least the SOC stock before and after in absolute terms (I would say that heathlands are much richer than your "Barren Land", no?).

Line 23-25, page 9: this is pretty well known pattern after afforestation. You can refer to fig. 1 in Goulden et al., 2011 (references at the end) (panel a), which by the way could be used as a function of production (and inputs, panlel c) for an interesting SOC modelling study of your data. Anyway, you could discuss these patterns.

Line 7, page 10: could you explain what is a "eratica"? If it is one identifiable organism put the taxonomic name, otherwise explain, I'm really not familiar with the term. If it is latin, plural of "erraticus", mind you it is with double "r" and it could still be a bit obscure to many since at leats in latin it generically means something like "things that go around. . ." and not necessarily living things. If it's a discipline-specific context you might need to clarify.

Line 20, 23, page 10: it is nothing too weird that you still have some C left. You can refer to the study by Barré et al., 2010, and following studies on the LTBF network for having a picture of SOC evolution in barren conditions. It takes several decades for the soil to lose the C, and several millennia to lose all of it (you can also accept the approximation of "stable" C pool of Barré, if you like, it is virtually correct at your time scales). It is nevertheless pretty interesting to me that the degradation is so faster in

the upper topsoil than the lower topsoil. . . the LTBF are cultivated in the 0-20, so this stratification is not observable. You might have there also some really interesting hints about the protection of SOC exerted by depth, maybe.

Line 30, page 10: definitely agree! But "bulk SOC stocks" is not necessarily 0-30. . . you could just use bulk SOC stocks in 0-5 cm, no? It seems you mean that bulk stocks in general are not to be used, like this.

Line 3, page 11: probably you mean the effects of the afforestation, rather than the afforestation itself

Line 6, page 11: in this case I would rather use a relative value for the delta, it's more immediate

Line 11, page 11: "false" is not the right term here. I mean it doesn't sound right in English. A statement can be false, using something cannot, no matter how badly you're using it that's not false. It can be misleading, for example, or other similar terms.

Line 16-17, page 11: really do you need 5 studies to say that you have higher C inputs if you have some plants compared to no plants? Just asking. . .maybe not, to me it sounds pretty obvious, although correct.

Line 20-23, page 11: why do you use a median? If the distribution I skewed, as I bet it is, do the comparisons one by one. . .and use statistical tests! I mean, assess the significance of you comparisons, comparing the mean possibly. Than maybe yu can use also some more exotic things like medians, if you really like to, but for sure use p values in your comparisons. Ah, then you compare Birchnat to Birch50, without stating any number. . .

Line 25, page 11: maybe hypothesize is better term than assume, here, or "one might hypothesize at first"

Line 25-30: the fact that you have more POM from birches but more C stocks from grassland should be related to the C found in the <63nm and HF. The fact that yiou think

that such C was already there due to remnants is an explanation for what you find in the paragraph above. You have more stabile SOC here (which is desirable) compared to birch plantation because that SOC was already there due to soil chacteristics, at least this is what you suggest (you should also discuss a bit other possibilities, since you cannot be sure, such as "does grassland put C in stable fractions faster than birch"? Maybe not, but you should discuss this).

Line 29, page 12: ok, but wasn't this belonging to the previous paragraph (ah, btw, they are paragraphs, not chapters)?

Line 30, page 12: after all the medians you showed, now I fear this median might be grouping different sites. Median between what? (and please remember my former comment about using statistical test, for which a mean might be easier. I know you might have skewed distributions, but it's pretty hard to deal with them... I appreciate your effort in this sense, but still you need to deal with statistics, an aggregate number itself has no real meaning without error and statistics)

Line 27-28, page 12: ok, but this is a problem of your setup. You did not do the oxidation, the last step of the fractionation, so you do not have information about the stability of the material. If you did, you could relate your results to the stabilization.

Line 2-4, page 13: as above, the Zimmermann fractionation (Zimmermann et al., 2007) is not only physical, but it includes a chemical oxidation exactly for this reason (ok, it is a rough indication, but still it is an indication of stabilization). You decided to skip this. Fine, but it is your decision, not a flaw in the method...

Line 26-28, page 13: these correlations are weak, you need to state also the p-value, I'd say. For a $r^2 > 0.8$ I wouldn't be so strict, but these are rather low.

Line line 13-14, page 13: with "undetermined" you mean that you did not find any correlation? Try to be clear about these things, this sounds like a euphemism.

Line 14-20, page 13: since this is a rather important part of your study, could you please

analyze it more in detail? You could test some regressions on the different groups you indicate, and give the results (and p-values!), and try to demonstrate your hypothesis with your data. It's an interesting hypothesis, and you should find some correlation… instead of writing that "it is undetermined" just try to determine that stabilization, that's your job as scientist after all.

Line 29, page 13: what do you mean with "continuous"? That value is also not normalized by time, I can't understand that adjective in such context. To me "continuous" could refer here to a rate of inputs that did not change over 15 and over 50 years, but this is a (cumulative, so integrated over time and not a rate) mass. And what that increase the same for all the stands?!? What do you mean 15 t C ha-1 between 15 and 50 years?

Line 7: that most of the SOC is in the <63nm fraction is expected, that's just how SOC humification/degradation works, and it does not indicate much else. How do you think it relates to what follows?

Line 14-15, page 13: you wanted to "evaluate the SOC sequestration potential of afforestation on severely degraded soils in southern Iceland." and your key message is a recommendation about caution in choosing the sampling depth for soil surveys?!? I think you should focus a bit more on your main aims, you have some information there. And try to be consistent with such aims, write down your hypotheses, test them (also statistically) and tell me more about how it went. I wouldn't use the last line for a recommendation that just points out some shortcomings of your study, actually.

**References**

Goulden ML, Mcmillan AMS, Winston GC, Rocha A V., Manies KL, Harden JW, et al. Patterns of NPP, GPP, respiration, and NEP during boreal forest succession. Glob Chang Biol. 2011;17: 855–871. doi:10.1111/j.1365-2486.2010.02274.x Barré P, Eglin T, Christensen B, Ciais P, Houot S, Kätterer T, et al. Quantifying and isolating stable soil organic carbon using long-term bare fallow experiments. Biogeosciences.

2010;7: 3839–3850. doi:10.5194/bg-7-3839-2010 Zimmermann M, Leifeld J, Schmidt MWI, Smith P, Fuhrer J. Measured soil organic matter fractions can be related to pools in the RothC model. Eur J Soil Sci. 2007;58: 658–667. doi:10.1111/j.1365-2389.2006.00855.x
* * *

---

## Referee Comment (RC2) · R. Qualls (Referee) · 5 Sep 2018

Review of Hunsiker et al. This manuscript describes a very interesting study of the accumulation of carbon, particle density fractions and the clay fraction that would be relevant to adsorption of carbon in volcanic soils. It would be relevant to the literature on soil development during primary succession on volcanic soils, and perhaps to secondary succession on volcanic soils. One thing that is unique is that unlike in many studies of soil development during succession, there is only one species of tree involved, with one "variable" removed (with the exception of the grassland which provides and interesting contrast with deposition of carbon at different depths.

As the authors acknowledge, there is unfortunately no "time zero" for the afforestation of the birch since the barren plots seem to have organic matter left from a previous era when it must have been vegetated, as indicated by C contents that are greater, even at depth than the young birch plots. Perhaps some initial state can be inferred by extrapolation to zero time in the birch time sequence. The methods used were very pertinent to a study of soil development on volcanic substrates. The analyses of allophane and Fe and Al oxyhydroxides are just what this reviewer used in comparable studies. The separation of carbon by density fractions are also what Sollins et al. (see reference below) recommended to monitor the deposition of root detritus vs. the adsorbed or occluded carbon that might be expected with allophane and Fe/Al oxyhydroxides interactions.

There are a few things that I might suggest could be made clearer to the readers. In the description of the history of the sites, I was not able to follow which plots actually used for the study were associated with each history. Perhaps it would help to have a table listing each group of plots (barren, planted birch, natural birch, grassland) and relevant elements of history (previous land use, eroded, volcanic desert, volcanic sand deposition, etc.). In many comparable studies of chronosquences, a key question is the degree to which all vegetation/age types originated from the same parent material. Obviously they are all of volcanic origin, but some had different histories and there is no true "initial state" since there appears to be a buried A horizon. Perhaps clarify the discussion as to which sites can be considered subsets of "vegetation/age" classes can be considered as having the same initial states that differ by age or vegetation.

Study design and replication. The following paragraph makes it difficult to figure out the experimental design and replication: "Each of the land cover types and age categories described above was represented by three test sites, resulting in a total of 21 sampling sites (Figure 1; E). . . . . . . . . . . . . .r. At each site, five soil pits were randomly placed. At the woody sites, sampling occurred within one half of the crown diameter of a dominant mountain birch (Betula pubescens Ehrh. ssp. czerepanovii) tree. The soil was sampled with a cylindric metal core (Eijkelkamp Soil & Water, Giesbeek) of 100 cm3 volume and

5 cm in diameter at given soil intervals (0- 5, 5-10, 10-20 and 20-30 cm). The five sub-samples per depth interval were immediately mixed in order to form one composite sample. Thus, each depth interval per category was represented by three composite samples (Figure 1), resulting in a total of 84 composite samples." It is difficult to figure out the experimental design from paragraph and figure (Figure 1) seems to have some contradictions. There were 5 pits in each site. Part of the problem is the use of the words "land cover types" and "sites". Many authors use "site" to indicate the "treatment" and "plot" as the unit that serves as a replicate. I realized these were not randomly allocated treatments, but the nomenclature is confusing making it difficult to tell that there are 3 replicates per "vegetation/age" class. What is "category" in "depth interval per category, is this the same as site? Could site be referred to as "plot"? In Figure 1, the map is useful. But, in the maps B, C, and D I do not see asterisks, triangles, etc. as it says in the caption. The list of sites, profiles, and composite samples is only confusing. Perhaps you could list "vegetation/age" classes, "number of plots or sites within each class", "subsamples composited within each plot"... to make the number of true replicates apparent.

In the discussion, there are a couple of very relevant references that are comparable in terms of (1) the rate of carbon accumulation over time on volcanic soils, (2) the development of allophane and iron and aluminum oxyhydroxides and the role of adsorption of carbon, and (3) the use of density fractionation to examine the role of association of C with volcanic minerals and its refractory nature. These are listed below: Sollins, P., Spycher, G., Topik, C., 1983. Processes of soil organic matter accretion at a mudflow chronosequence, Mt Shasta, California. Ecology 64, 1273– 1282.

Lilienfein*, J., Qualls, R.G, Uselman*, S.M, and Bridgham S.D. 2003. Soil formation and organic matter accretion in a young andesitic chronosequence at Mt. Shasta, California. Geoderma 116:249-264.

Lilienfein J, Qualls R.G, Uselman* S.M.and Bridgham S.D. 2004. Adsorption of dissolved organic carbon and nitrogen in soils of a weathering chronosequence. Soil

Science Society of America Journal 68 292-305.

Other specific notes are listed below:

Abstract lines 26 through 29. The cause and effect does not seem clear. Suggested revision: "After 50 years of birch growth, the SOC stock is lower than that of a naturally growing birch woodland. Suggesting that afforested stands could sequester additional SOC beyond 50 years of growth."

please spell out sodium polytungstate

Page 14, lines 14-15 needs rewriting.

Robert G. Qualls

---

## Author Comment (AC2) · 26 Dec 2018

Dear Robert Qualls

we want to thank you for your valuable review. Some of your remarks concerning the statistics and the topic of the essential status t0 for chronosequence studies were already key points which we also discussed during the data analysis process and the writing of this manuscript. Due to the small sub-datasets per age and depth class and the not-normal distribution of these sub-datasets, we decided to apply the Wilcoxon rank-sum test (WRS), which is applied if the statistical requirements for T-test are not given. Further responses on this topic are listed below.

[Figure]

At this point I also apologize that we have not earlier answered to your reviews. The reason is that this manuscript records one part of my PhD studies and I, as main author of the manuscript, have not been employed at any research institute for more than 1.5 years. Since that time, I work at an enterprise in the private industry. This is not an excuse to ignore your review comments, but the time and software recourses are very limited or no longer available. Nevertheless, we modified our manuscript according to your comments as thoroughly as possible, constructed replies to your comments and submit the revised manuscript and the answer of the authors hereby.

Kind regards Matthias Hunziker, main author.

Referee 2; Robert Qualls

General comment This manuscript describes a very interesting study of the accumulation of carbon, particle density fractions and the clay fraction that would be relevant to adsorption of carbon in volcanic soils. It would be relevant to the literature on soil development during primary succession on volcanic soils, and perhaps to secondary succession on volcanic soils. One thing that is unique is that unlike in many studies of soil development during succession, there is only one species of tree involved, with one "variable" removed (with the exception of the grassland which provides and interesting contrast with deposition of carbon at different depths. As the authors acknowledge, there is unfortunately no "time zero" for the afforestation of the birch since the barren plots seem to have organic matter left from a previous era when it must have been vegetated, as indicated by C contents that are greater, even at depth than the young birch plots. Perhaps some initial state can be inferred by extrapolation to zero time in the birch time sequence. The methods used were very pertinent to a study of soil development on volcanic substrates. The analyses of allophane and Fe and Al oxyhydroxides are just what this reviewer used in comparable studies. The separation of carbon by density fractions are also what Sollins et al. (see reference below) recommended to monitor the deposition of root detritus vs. the adsorbed or occluded carbon that might be expected with allophane and Fe/Al oxyhydroxides interactions.

Author's answer An exponential function based on the time-dependent SOC stocks (0-30 cm) of B15-B50 as input data (computed in Excel) showed an SOC stock as initial status (t0) of 26.25 t C ha-1 (y=26.246e0.0111x, R2=0.44). This is a quite smaller SOC stock value than found at Barren Land (39 t C ha-1). According to these, it seems that at the sites of B15, B20, B25 and B50 the initial SOC stock before any afforestation activities starts is distinct lower than the used initial status of severely degraded land (Barren Land) in the present study.

Revised version The authors suggest to include the reviewer's input by inserting this finding in the section concerning the SOC stock (0-30 cm) similarly.

Comment 1 There are a few things that I might suggest could be made clearer to the readers. In the description of the history of the sites, I was not able to follow which plots actually used for the study were associated with each history. Perhaps it would help to have a table listing each group of plots (barren, planted birch, natural birch, grassland) and relevant elements of history (previous land use, eroded, volcanic desert, volcanic sand deposition, etc.). In many comparable studies of chronosquences, a key question is the degree to which all vegetation/age types originated from the same parent material. Obviously they are all of volcanic origin, but some had different histories and there is no true "initial state" since there appears to be a buried A horizon.

Author's answer: In our opinion, the description is good enough and a table would overload the section which already contains Figure 1 about the location and the setup of the soil sampling. In the revised version, we labeled the different tested categories.

Revised version: However, we can create a short table in the revised version of the manuscript.

Comment 2 Perhaps clarify the discussion as to which sites can be considered subsets of "vegetation/age" classes can be considered as having the same initial states that differ by age or vegetation.

Author's answer: In the discussion, we considered this comment.

Revised version: The results indicate that spatial variability must be taken into account when analyzing SOC of volcanic soils, especially when deeper than 10 cm, between the sampled sites and the land cover categories (i.e. grassland, barren, etc). This is even more relevant in landscapes with past or recent erosion processes as soil forming process. Thus, the equality or comparability of the sites, except for the studied variable, is not ensured for space-for-time substitution sampling approaches under such circumstances as performed in the present study (Walker et al., 2010). Hence, it is misleading to use the selected Barren Land sites, which were selected at 4 km distance from the afforested sites (Birch15, Birch20, Birch25 and Birch50) and 15 km from Birchnat, as initial status (t0) for discussing the effect of afforestation and calculating any SOC sequestration rates.

Comment 3 Study design and replication. The following paragraph makes it difficult to figure out the experimental design and replication: "Each of the land cover types and age categories described above was represented by three test sites, resulting in a total of 21 sampling sites (Figure 1; E). . . .. . .. . .. . .. . ..r. At each site, five soil pits were randomly placed. At the woody sites, sampling occurred within one half of the crown diameter of a dominant mountain birch (Betula pubescens Ehrh. ssp. czerepanovii) tree. The soil was sampled with a cylindric metal core (Eijkelkamp Soil & Water, Giesbeek) of 100 cm3 volume and 5 cm in diameter at given soil intervals (0- 5, 5-10, 10-20 and 20-30 cm). The five subsamples per depth interval were immediately mixed in order to form one composite sample. Thus, each depth interval per category was represented by three composite samples (Figure 1), resulting in a total of 84 composite samples."

It is difficult to figure out the experimental design from paragraph and figure (Figure 1) seems to have some contradictions. There were 5 pits in each site. Part of the problem is the use of the words "land cover types" and "sites". Many authors use "site" to indicate the "treatment" and "plot" as the unit that serves as a replicate. I realized these were not randomly allocated treatments, but the nomenclature is confusing making it

difficult to tell that there are 3 replicates per "vegetation/age" class. What is "category" in "depth interval per category, is this the same as site? Could site be referred to as "plot"?

Author's answer: We see the problem which is mentioned by the reviewer. During the writing of the manuscript we intensively thought about the most appropriate terminology. Throughout the manuscript, we keep the terminology constant. It is correct that there were 5 pits per site. "Category" in "depth interval per category" is land cover category in combination with the age of vegetation growth e.g. "Barren Land", "Grass50", "Birch15". And the term "category" is not the same as "site" because we tested three sites per category. In our study setup the term "site" is referred to as "plot" which serves as a replicate according to the reviewer.

Revised version: Each of the land cover types (e.g. Barren Land, afforested birch stands) and age categories (e.g. 15, 20, 50 yrs old birch stand) described above was represented by three test sites (3 replicates) (Figure 1; E).

Thus, each depth interval per category was represented by three composite samples (3 replicates per depth interval) (Figure 1), resulting in a dataset of total of 84 composite samples.

Comment 4 In Figure 1, the map is useful. But, in the maps B, C, and D I do not see asterisks, triangles, etc. as it says in the caption. The list of sites, profiles, and composite samples is only confusing. Perhaps you could list "vegetation/age" classes, "number of plots or sites within each class", "subsamples composited within each plot". . . to make the number of true replicates apparent.

Author's answer: The points of the test sites were categorized as it is mentioned in the caption. However, we keep the list with the numbers of test sites, soil pits, collected samples and composite samples.

Revised version: Figure 1 was changed.

Comment 5 In the discussion, there are a couple of very relevant references that are comparable in terms of (1) the rate of carbon accumulation over time on volcanic soils, (2) the development of allophane and iron and aluminum oxyhydroxides and the role of adsorption of carbon, and (3) the use of density fractionation to examine the role of association of C with volcanic minerals and its refractory nature.

Author's answer: As I mentioned in the introduction of the author's, due to the change of the workplace I have no longer access to the scientific literature and a request for renewing my university account was declined. Nevertheless, Sollins et al. 1983 is cited in the revised version of the manuscript.

These are listed below: Sollins, P., Spycher, G., Topik, C., 1983. Processes of soil organic matter accretion at a mudflow chronosequence, Mt Shasta, California. Ecology 64, 1273– 1282.

Lilienfein*, J., Qualls, R.G, Uselman*, S.M, and Bridgham S.D. 2003. Soil formation and organic matter accretion in a young andesitic chronosequence at Mt. Shasta, California. Geoderma 116:249-264.

Lilienfein J, Qualls R.G, Uselman* S.M.and Bridgham S.D. 2004. Adsorption of dissolved organic carbon and nitrogen in soils of a weathering chronosequence. Soil Science Society of America Journal 68 292-305.

Revised version: Our findings are confirmed by the results of Sollins et al., (1983), who studied C dynamics at four mudflow chronosequences at Mt. Shasta in California and hence stated that the heavy fraction is an important C sink (37-72% of total C).

At these sites, the SOC stock (Figure 2) consisted mostly of carbon which was stored in the '< 63 $\mu$m' (65 %) and HF (28 %) fractions, respectively (Table 3) which is in accordance with Sollins et al., (1983).

Specific comments: Comment 6 Abstract lines 26 through 29. The cause and effect does not seem clear. Suggested revision: "After 50 years of birch growth, the SOC

stock is lower than that of a naturally growing birch woodland. Suggesting that afforested stands could sequester additional SOC beyond 50 years of growth."

Author's answer: Reviewer's suggestion was accepted.

Revised version: After 50 years of birch growth, the SOC stock is lower than that of naturally growing birch woodland. Hence, afforested stands can sequester additional SOC after 50 years of birch growth.

Comment 7 please spell out sodium polytungstate

Author's answer: The suggestion was accepted and the sentence was changed.

Revised version: The particulate organic material (POM) was separated from the denser organic material in the mineral-associated sand and aggregate fraction (heavy fraction; HF) by density fractionation (1.8 g cm-3, sodium polytungstate from Sometu) on the soil material (> 63 microns).

Comment 8 Page 14, lines 14-15 needs rewriting.

Author's answer: The sentence was changed.

Revised version: The pattern that the upper most sampling intervals of the vegetated sites (dotted circle) are decoupled from the nested scatters, was also observed at the relationship between the selected SOC pools (bulk SOC concentration, < 63 $\mu$m SOC concentration) and the organo-mineral complexes (Figure 5; E, F).

Basel, 26 December 2018, M. Hunziker

---

## Author Response (AR1)

Dear Lorenzo Menichetti

we want to thank you for your valuable reviews. Some of your remarks concerning the statistics and the topic of the essential status t0 for chronosequence studies were already key points which we also discussed during the data analysis process and the writing of this manuscript. Due to the small sub-datasets per age and depth class and the not-normal distribution of these sub-datasets, we decided to apply the Wilcoxon rank-sum test (WRS), which is applied if the statistical requirements for T-test are not given.

At this point I also apologize that we have not earlier answered to your reviews. The reason is that this manuscript records one part of my PhD studies and I, as main author of the manuscript, have not been employed at any research institute for more than 1.5 years. Since that time, I work at an enterprise in the private industry. This is not an excuse to ignore your review comments, but the time and software recourses are very limited or no longer available. Nevertheless, we modified our manuscript according to your comments as thoroughly as possible, constructed replies to your comments and submit the revised manuscript and the answer of the authors hereby.

Kind regards

Matthias Hunziker, main author.

Author's answers to the comments of the two referees

Referee 1; Lorenzo Menichetti

**General comment**

The study is conducted in one of the most fascinating setups ever (and this means something to me, I have to admit my bias here. I am pretty interested in these data also because of a fascination for Iceland on one side and for Andosols on another side). It also contains some pretty interesting data, and their impact can be substantial.

There is a lot of work behind the data, and there are a lot of points and really long-term treatments (also extremely interesting treatments by the way! I would personally want to know more about the Barren Land treatment, but the experimental design seems in general fantastic, with really long treatments although with the problem of underlying soil variability). There seems to be a lot of valuable information here. The manuscript nevertheless falls shortly in making the most out of such data. It lacks statistical tests, and hypotheses are not discussed in a testing framework. You could and should work more on it, in my opinion.

You should restructure a bit your hypotheses and conclusions. You mention in the text a bit too many times the same things, that SOC below 10 cm was surprising, it might be due to soil characteristic and it was due (you suppose) to previous accumulation and fossil soil layers. It all makes sense to me, and I understand this was unexpected for you, but try to make some logical blocks for it.

After you cleared the hypothesis testing framework, then you can work on the statistics for comparison, and clarify each comparison. Compare something to something else, always. If something increases, always states compared to what it increases. And put there the results from a statistical test for the comparison. You do not need to use necessarily fancy tests, just the basics t-test, ANOVA, linear regressions, combined in different ways, could be enough as tools. But use them extensively and ideally do not state anything without being able to offer some statistical evidence (there are of course exceptions, but in your case they look more like the rule).

As a suggestion for future developments, I think you should think to some more detailed modelling, at least with some compartmental model with analytically solved steady state, and try to model the input functions (I attached a reference that might be useful as a rough start). If you get right the function of input variation, a SOC model can be calibrated on your data and give you the change of the steady state. . . and predict the future steady state, meaning the C you could accumulate in these soils. But this is clearly outside the scope of your present study, I just got excited about the idea. . . and to me such idea is exemplifying why I do believe you have a lot of useful information here.

**Specific comments:**

**Comment 1**

Abstract: The idea of C "sequestered as labile" sounds a bit counterintuitive in itself to me... I'm not sure I would consider labile C as "sequestered", since it stays for very short time anyway. But this is philosophy after all, not a major comment.

**Author's answer:**

We changed the sentence to:

**Revised version:**

10 Therefore and due to absence of any increase in the tested mineral-associated SOC fractions, we assume that the afforestation process evokes a carbon deposition in the labile SOC pools. Consequently, parts of this plant-derived, labile SOC may be partly released to the atmosphere during the process of stabilization with the mineral soil phases in the future.

15 **Comment 2:**

Line 11, page 3: Define "sequestration rate" better. The inputs in an afforestation follow a function variable over time, and this causes a continuous variation in the rate of change of C stored. I am sure this is what you meant, but "sequestration rate" was not defined before and it might be ambiguous. Is the "sequestration factor", right? Yes, that would be variable over time following the function of input variation.

**Author's answer:**

We agreed and changed the sentence to:

**Revised version:**

25 The establishment of a vegetation community passes through different development stages, consequently, the sequestration rate, as a function of SOC change over time, is not linear until the new SOC stock equilibrium is reached (Smith et al., 1997; Six et al., 2002; Stewart et al., 2007).

**Comment 3:**

Line 2, page 6: you do not complete the Zimmermann fractionation! You skip the last step, the oxidation, right? You need to mention this in M&M. This is pretty crucial to understand many of your following statements (I initially missed this detail and I read almost 2/3rd of the manuscript before realizing it).

**Author's answer:**

In the submitted version we already mentioned that we skipped the oxidation method ("Compared to Zimmermann et al., (2007), the present study did not conduct oxidation with sodium hypochlorite (NaOCl) to determine the resistant SOC pool.").

But we added in the revised version the following sentence:

**Revised version:**

Hence, the present study did not measured the SOC in the NaOCl resistant fraction (rSOC).

**Comment 4:**

Line 5-6, page 7: I guess you mean "each soil fraction", giving an example for the fine soil fraction only, right? But the equation was the same for all fractions, right?

**Author's answer:**

Fine soil fraction is not used in the term of any SOC fraction. It is used as a size-dependent soil material fraction. Commonly fine soil material is defined as the soil material which is smaller than 2mm in diameter and is determined by sieving with a 2mm sieve.

The equation given in line 8 on page 7 was used to calculate the bulk SOC stocks based on the variables SOC concentration, bulk density of the fine earth material (< 2mm) and the sampled soil depth. The bulk density of the fine earth material (<2mm) was determined by dry sieving and water displacement of the coarse material (> 2mm).

**Revised version:**

Hence, the study calculated the amount of soil organic carbon which was stored in the fine soil fraction (< 2 mm) within the top 30 cm.

**Comment 5:**

Line 28, page 7: this is confusing. You sampled also "Barren Land", no? Which had a high ferrhydrite content, as you say before. High compared to what? Do you mean that ferrhydrite was "high", but allophane material was higher than this latter? Or do you mean "all other samples than Barren Land"?

5 **Author's answer:**

We mean that the highest clay concentrations were found at Barren Land. Further the concentrations at Barren Land are also higher than given in the literature for desert Vitrisols.

The passage was changed and shortened.

10 **Revised version:**

At Barren Land, we found the highest concentrations of allophane and ferrihydrite clay minerals (Table 2). These high concentrations stand in contrast to the typically low concentration (2-5 %) found in desert Vitrisol (Arnalds, 2015d). Lilienfein et al., (2003) found an increase of allophane and ferrihydrite concentrations with increasing age of mudflow soils at Mt. Shasta. Based on these findings, our results indicate that the soils at Barren Land are pedogenetically developed and

15 the high carbon and clay contents found on Barren Land are more representative of severely degraded soils than Icelandic desert soils.

**Comment 6:**

20 Line 1, page 8: what do you mean with "nutrient contents"? Other nutrients than C? Please specify which nutrients. And in Table 1 you report only the C:N but not N, it is quite difficult to read this statement in the data (I would need to extract the values and do the calculation from C and C:N ratio). If also N was important, please report it with direct measurements, otherwise talk only about C.

25 **Author's answer:**

The authors agreed and changed the sentence to:

**Revised version:**

SOC concentrations varied between 0.6 and 9.8% within the whole dataset of the 84 samples (Table 1).

**Comment 7:**

Line 13, page 8: maybe you cannot generalize so much. Only the degraded volcanic soil soils you sampled showed that, I would say you cannot say the same in general.

**Author's answer:**

The authors agreed and rewrote the sentence as:

**Revised version:**

Based on the analysis of the C concentrations of the soil samples, the study showed that the un-vegetated, severely degraded volcanic soils contained appreciable amounts of SOC. And further, afforestation with mountain birch increased the soil C concentration during the first 50 years of shrub establishment, predominantly in the top 10 cm.

**Comment 8:**

line 26, page 8: "usually tested" is generic term that has pretty much no meaning. By whom? Maybe my usual tests are not the same, maybe I am personally used to something else?

**Author's answer:**

The title was changed to

**Revised version:**

Afforestation seems to increase the SOC stock in the top 30 cm

**Comment 9:**

Line 3-13, page 9: all this discussion has the defect of not considering the starting point. Soil C stocks are an equilibrium defined over several decades/centuries by the inputs. A land use change represents a change from a possible equilibrium state to something else, in theory a new equilibrium after climax. If this means a loss or a gain of SOC depends on the new inputs (so the age of the plantation in this case), but also on what was there before. In all these comparisons you should indicate at least the SOC stock before and after in absolute terms (I would say that heathlands are much richer than your "Barren Land", no?).

**Author's answer:**

The paragraph which was mentioned by the referee contains a review of already published results by different authors. We included the demanded information taken from the literature where it was available. We added the age of the tested sites as far it was given in the cited literature.

5   **Revised version:**

In the present study, the initial state before afforestation starts is represented by the sites of Barren Land. The SOC stock (0-30 cm) at Barren Land (median value: 39 t C ha-1) is higher (p > 0.05) than the SOC stocks at Birch15, Birch20 and Birch25. The present study found a continuous increase in median SOC stock (0-30 cm) with birch stand age (Birch15: 31; Birch20: 33; Birch25: 36; Birch50: 46 t C ha-1) (Figure 2). After 50 years of birch growth the SOC stock (0-30 cm) of

10  Birch50 is significantly (p = 0.05) higher than the SOC stocks of younger birch stands (Birch15, Birch20) or severely degraded soil (Barren Land) (Figure 2). The given results of the SOC stocks (0-30 cm) might lead to the assumption that the soil acts as C source during the first 25 years of the establishment of birch and that there is a carbon sink between after 25 years until 50 years of birch growth. This would be in accordance with Hunziker et al., (2017) who found a decline of the SOC stock (0-30 cm) during the first 40 years of green alder encroachment on former subalpine pastures. Another finding of

15  the present study is that after 50 years of birch growth, the SOC stock was still significantly (p = 0.05) lower than that of the old-growth woodlands of Birchnat (Δ 15 t ha-1) (Figure 2). This means that the soils at Birch50 can sequester additional organic carbon during the succession towards mature woodlands which reflects the equilibrium state. Overall, the results indicate that afforestation by mountain birch, and the establishment of birch woodlands, can significantly increase the SOC stock (0-30 cm) (Birch15-Birch50), which is in accordance with Icelandic studies given in the literature.

20  During the period between Birch15 and Birch50 (35 years), the sequestration rate is 0.42 t C ha-1 a-1 on average, without taking the SOC stock of Barren Land (as status before afforestation begins) as reference for calculation. The reason for this is given in the assumption that Barren Land contains a lot of SOC which is not originated from the revegetation process. The rate is lower than the given removal factor of 0.51 t C ha-1 a-1 for afforestation activities (Hellsing et al., 2016).

A literature review revealed that the succession of already vegetated heathland to birch woodland in eastern Iceland shows

25  no change in C stocks (Ritter 2007). The SOC stocks were about 40 t C ha-1 (0-20 cm) for 26 and 97 year old birch stands. Snorrason et al., (2002) found a higher SOC stock (0-30 cm) in a 54 year old birch stand (65 t C ha-1), compared to that of grassland (54 t C ha-1) at Gunnarsholt, which leads to the assumption that the effect of afforestation is more effective than that of revegetation concerning SOC sequestration. However, Snorrason et al., (2002) and Ritter (2007) did not reported the SOC stock of the initial status before ecosystem change began. Soil development and natural vegetation succession on

30  moraine till after glacial retreat is another typical process of land cover change in Iceland. Vilmundardóttir et al., (2015) found a SOC accumulation within the top 20 cm from 0.9 (initial status) and 13.5 t C ha-1 at sites with a maximum age of 120 years, thereby demonstrating that the process of vegetation succession on moraine till leads to an increase in soil carbon stock. Our results indicate that the change in SOC stocks during afforestation with mountain birch on severely degraded soils (Figure 2) is comparable with those given for shrub encroachment in the cited literature.

**Comment 10:**

Line 23-25, page 9: this is pretty well known pattern after afforestation. You can refer to fig. 1 in Goulden et al., 2011

5 (references at the end) (panel a), which by the way could be used as a function of production (and inputs, panlel c) for an interesting SOC modelling study of your data. Anyway, you could discuss these patterns.

**Author's answer:**

In our opinion, the main reasons for the SOC stock patterns (decrease and after increase) is not mainly due to the change of

10 the ecosystem production and the delivery of organic material to the soil. The main reason is that Barren Land which was assumed to be t0, can be badly taken as t0 status.

Therefore, we decided to apply a depth-dependent SOC fractionation and also a physical SOC fractionation to characterize the SOC patterns during the establishment of mountain birch woodlands on severely degraded volcanic soils. We referred to Goulden et al. 2011 by knowing that Goulden et al. (2011) did measure the C-stock of the forest floor and not the SOC stock

15 of the mineral soil phases.

**Revised version:**

The results of the SOC stocks within the commonly used soil depths of 30 cm of the tested categories indicate that the SOC pool decreases between Barren Land and 15 years old birch stands. It then increases during tree establishment to reach the

20 level of naturally grown birch woodlands (Figure 2). This pattern is comparable with other field studies (e.g. Goulden et al., 2011; Hunziker et al., 2017). However, the analysis of mineral SOC dynamics in such a temporally dynamic landscape, which results in unequal SOC and volcanic clay concentrations patterns across the tested categories, calls for more detailed and alternative methods (Table 1, Table 2). Thus, the present study further focused on the vertical distribution of the SOC and its quality, to verify whether afforestation results in the soil becoming a C source, and whether more C is sequestered

25 during revegetation than afforestation.

**Comment 11:**

Line 7, page 10: could you explain what is a "eratica"? If it is one identifiable organism put the taxonomic name, otherwise explain, I'm really not familiar with the term. If it is latin, plural of "erraticus", mind you it is with double "r" and it could still be a bit obscure to many since at leats in latin it generically means something like "things that go around. . ." and not

5   necessarily living things. If it's a discipline-specific context you might need to clarify.

**Author's answer:**

In this term it means the material of volcanic eruptions.

10   **Revised version:**

However, this is a typical pattern of volcanic soils which are also characterized by biologicaly active soil layers buried by ash from volcanic eruptions.

15   **Comment 12:**

Line 20, 23, page 10: it is nothing too weird that you still have some C left. You can efer to the study by Barré et al., 2010, and following studies on the LTBF network for having a picture of SOC evolution in barren conditions. It takes several decades for the soil to lose the C, and several millennia to lose all of it (you can also accept the approximation of "stable" C pool of Barré, if you like, it is virtually correct at your time scales). It is nevertheless pretty interesting to me that the

20   degradation is so faster in the upper topsoil than the lower topsoil. . . the LTBF are cultivated in the 0-20, so this stratification is not observable. You might have there also some really interesting hints about the protection of SOC exerted by depth, maybe.

**Author's answer:**

25   It is right that it is quite important to recognize that it can take a long time for soils of collapsed barren ecosystems to emit all the carbon from the soils to the atmosphere, and we stress this point in the paper. We include this citation for further stressing the point.

**Revised version:**

This implies that the soils of Barren Land contain a certain amount of SOC due to earlier soil formation processes prior to disturbance and SOC accumulation, and which occurred before the soil profile was truncated by soil erosion processes. Such pedologic conditions can ask for soil C decay studies, as it was introduced by Barré et al., (2010). However, this was not the objective of the present study.

**Comment 13:**

Line 30, page 10: definitely agree! But "bulk SOC stocks" is not necessarily 0-30. . . you could just use bulk SOC stocks in 0-5 cm, no? It seems you mean that bulk stocks in general are not to be used, like this.

**Author's answer:**

We used the term "bulk SOC stocks" for the unfractionated SOC stocks as given in Figure 2. However, we deleted "bulk" throughout the manuscript or changed it to "unfractionated".

**Revised version:**

The subdivision of the studied soil columns of 30 cm in four sampling intervals explains the higher SOC stocks at Barren Land and Grass50. This is due to the higher values in the intervals "10-20 cm" and "20-30 cm" compared to the afforested birch sites (Birch15-Birch50), which constitute older buried soils (Table 1, Figure 2). The subdivision further characterizes the patterns found of SOC stock (0-30 cm) (chapter 3.2), with the high SOC stocks (0-30 cm) at Barren Land and Grass50 being caused by the carbon pool located deeper than 10 cm soil depth. Under the given site conditions, it is questionable to apply the commonly used soil depth of 30 cm for SOC stock monitoring (Aalde et al., 2006), to sampled SOC that originates from buried soils, as it distorts the effects of restoration activities in the results of SOC concentration and SOC stock. Based on this understanding, the SOC stocks (0-30 cm) do not reveal that afforestation caused a C loss during the first 25 years of mountain birch establishment at such severely degraded sites, and that the effects of revegetation is more effective than those of afforestation by mountain birch within the first 50 years.

**Comment 14:**

Line 3, page 11: probably you mean the effects of the afforestation, rather than the afforestation itself

**Author's answer:**

The sentence was changed to:

**Revised version:**

Based on this understanding, the SOC stocks (0-30 cm) do not reveal that afforestation caused a C loss during the first 25 years of mountain birch establishment at such severely degraded sites, and that the effects of revegetation is more effective than those of afforestation by mountain birch within the first 50 years.

**Comment 15:**

Line 6, page 11: in this case I would rather use a relative value for the delta, it's more immediate

**Author's answer:**

We added the relative value.

**Revised version:**

However, the SOC stock (0-10 cm) of 50-year old birch woodlands is still lower ($\Delta$ 5 t C ha-1; 16 %) than the stocks identified at the Birchnat sites.

**Comment 16:**

Line 11, page 11: "false" is not the right term here. I mean it doesn't sound right in English. A statement can be false, using something cannot, no matter how badly you're using it that's not false. It can be misleading, for example, or other similar terms.

**Author's answer:**

We changed to word to "misleading"

**Revised version:**

Hence, it is misleading to use the selected Barren Land site as initial status (t0) for discussing the effect of afforestation and calculating any SOC sequestration rates.

**Comment 17:**

Line 16-17, page 11: really do you need 5 studies to say that you have higher C inputs if you have some plants compared to no plants? Just asking. . .maybe not, to me it sounds pretty obvious, although correct.

10 **Author's answer:**

We changed the references.

**Revised version:**

The net primary production (NPP) of a landscape is increased during afforestation. Hence, the supply of organic material to
15 the soil is higher at shrubby sites compared to barren areas (e.g. Bjarnadottir et al., 2007).

**Comment 18:**

Line 20-23, page 11: why do you use a median? If the distribution I skewed, as I bet it is, do the comparisons one by one. .
20 .and use statistical tests! I mean, assess the significance of you comparisons, comparing the mean possibly. Than maybe yu can use also some more exotic things like medians, if you really like to, but for sure use p values in your comparisons. Ah, then you compare Birchnat to Birch50, without stating any number. . .

**Author's answer:**
25 The sub-datasets (per age class and depth interval) consists of 3 replicates. We looked for a significance test for subsets with not-normal distributed data and only three samples. We applied the Mann-Whitney U Test. With this test, the lowest p-value we got was 0.05 hence the p-value was in any of the tested cases not lower than 0.05. Due to ending of the SPSS license, unfortunately I only have the p-values for the SOC stock results (Figure 2). However, we include this data in the revised manuscript for the results concerning the SOC stock data.
30 Further, we decided to apply descriptive statistical methods to describe the patterns. In our opinion and due to a study setup with only 3 replicates, using the median value instead of the mean value is more reasonable.

As I mentioned in the introduction of the author's, doing additional statistical tests would be way beyond the practicality of such endeavor due to change of workplace. If I would repeat this study or establish another study setup, I would reduce the number of strata and increase the number of samples/replicate per strata to n=5.

**Revised version:**

The revised version of the manuscript contains labels for significant differences per depth interval in Figure 2 (SOC stocks 0-30cm) and also the p-values in the text concerning the SOC stocks in any case of significance.

Revised version formerly Lines 20-23, page 11:

The sites at Birchnat contained 95 mg POM g-1 soil. The lower value at Birchnat (95 mg POM g-1 soil) compared to Birch50 (174 mg POM g-1 soil) can be explained by the lower productivity of Birchnat due to the already undergone self-thinning process during the forest development at Birchnat.

**Comment 19:**

Line 25, page 11: maybe hypothesize is better term than assume, here, or "one might hypothesize at first"

**Author's answer:**

We agreed

**Revised version:**

According to these results, it is hypothesized firstly that afforestation is a more effective restoration process than revegetation with grasses, in terms of supplying organic material, and hence carbon to the soil phases and secondly, this supply increases exponentially during the establishment of afforested birch woodlands.

**Comment 20:**

Line 25-30: the fact that you have more POM from birches but more C stocks from grassland should be related to the C found in the <63nm and HF. The fact that yiou think that such C was already there due to remnants is an explanation for what you find in the paragraph above. You have more stabile SOC here (which is desirable) compared to birch plantation because that SOC was already there due to soil chacteristics, at least this is what you suggest (you should also discuss a bit other possibilities, since you cannot be sure, such as "does grassland put C in stable fractions faster than birch"? Maybe not, but you should discuss this).

**Author's answer:**

In this paragraph, we present the mass of the POM material, which was found after the wet sieving procedure and the density fractionation (> 63 µm and < 1.8 g cm-3). As it is written, we use it as an indicator for the supply of organic material and carbon into the soil.

**Revised version:**

No changes were made.

**Comment 21:**

Line 29, page 12: ok, but wasn't this belonging to the previous paragraph (ah, btw, they are paragraphs, not chapters)?

**Author's answer:**

In our opinion, we don't see any conflict. We therefore did not change the manuscript according to this comment.

**Revised version:**

No changes were made.

**Comment 22:**

Line 30, page 12: after all the medians you showed, now I fear this median might be grouping different sites. Median between what? (and please remember my former comment about using statistical test, for which a mean might be easier. I know you might have skewed distributions, but it's pretty hard to deal with them. . . I appreciate your effort in this sense, but still you need to deal with statistics, an aggregate number itself has no real meaning without error and statistics)

**Author's answer:**

We changed the sentence in accordance to better understanding.

In our opinion and due to a study setup with only 3 replicates, using the median value instead of the mean value is more reasonable. As I mentioned in the introduction of the author's, doing additional statistical tests would be way beyond the practicality of such endeavour due to change of workplace.

For calculating a mean value of values from a dataset, the values and its distribution need to fulfill some requirements. Due to the small subdatasets in our study and distribution of the data we decided to not calculate the mean value and to show the median value.

If I would repeat this study or establish another study setup, I would reduce the number of strata and increase the number of samples/replicate per strata to n=5.

**Revised version:**

During afforestation, the increases between the median C stocks of the POM and '< 63 μm' fractions were 8 (+163 %) (p = 0.05) and 6 (+34 %) t C ha-1 between Birch15 and Birch50, while the SOC stock of the HF fraction seemed to stagnate at about 9 t C ha-1 during the same observation time (Table 3).These increases are explained by the increases of the '< 63 μm'-C and the POM-C concentrations during the afforested time span (Figure 3, Figure 4).

**Comment 23:**

Line 27-28, page 12: ok, but this is a problem of your setup. You did not do the oxidation, the last step of the fractionation, so you do not have information about the stability of the material. If you did, you could relate your results to the stabilization.

**Author's answer:**

It is true that we did not the chemical fractionation by oxidation as it is mentioned in Zimmermann et al. 2007 for all 84 "<63μm" samples. We tested the wet-oxidation with NaOCl on 42 samples (<63μm). However, the need of time and chemicals was out of scale with regard to the output due to the mineralogy of the volcanic soil material.

Another reason is that according to Jagadamma et al., 2010 and Lutfalla et al., 2014, it is questionable whether the wet-oxidation by NaOCl is the proper method to quantify the resistant SOC (rSOC).

Further, Zimmermann et al., (2007) did not analyze volcanic soils by the NaOCl-oxidation method. Other tests with volcanic soils from Iceland showed that oxidation with H2O2 do not oxidize the materials due to e.g. the Mn minerals.

Based on these reasons, we, however, decided to skip the oxidation step.

The sentences were rewritten and shortened.

**Revised version:**

This result can be attributed to a stabilization of the SOC due to its binding with the colloid fraction, which contains clay-sized minerals and organo-mineral complexes. However, the extraction of the material of the '< 63 μm' fraction by the physical separation technique of Zimmermann et al., (2007) due to the mineralogy of the samples. Hence, the chosen method in this study does not give information about the location of the organic matter in the '< 63 μm' fraction and consequently, the degree of the SOC stabilization.

**Comment 24:**

Line 2-4, page 13: as above, the Zimmermann fractionation (Zimmermann et al., 2007) is not only physical, but it includes a chemical oxidation exactly for this reason (ok, it is a rough indication, but still it is an indication of stabilization). You decided to skip this. Fine, but it is your decision, not a flaw in the method. .

**Author's answer:**

See comment above.

**Revised version:**

No changes were made.

Comment 25:

Line 26-28, page 13: these correlations are weak, you need to state also the p-value, I'd say. For a r^2>0.8 I wouldn't be so strict, but these are rather low.

**Author's answer:**

The correlations were computed in excel and excel does not give any p-values for correlation tests.

As I mentioned in the introduction of the author's, doing additional statistical tests would be way beyond the practicality of such endeavour due to change of workplace.

**Revised version:**

No revision on this comment.

**Comment 26:**

Line line 13-14, page 13: with "undetermined" you mean that you did not find any correlation? Try to be clear about these things, this sounds like a euphemism.

**Author's answer:**

We are agree and changed the sentence.

**Revised version:**

The stabilization of the SOC in the form of metal-humus complexes seems to be hampered due to the relatively high measured pH-values (Table 1), which were higher than the upper threshold value of 5.0 for the building of metal-humus complexes given in the literature (Figure 5; D, F).

**Comment 27:**

Line 14-20, page 13: since this is a rather important part of your study, could you please analyze it more in detail? You could test some regressions on the different groups you indicate, and give the results (and p-values!), and try to demonstrate your hypothesis with your data. It's an interesting hypothesis, and you should find some correlation. . . instead of writing that "it is undetermined" just try to determine that stabilization, that's your job as scientist after all.

**Author's answer:**

Regression analysis (R- values, R2-values and p-values were performed in Excel with the data analysis toolbox.

**Revised version:**

5   P-values were added in section 3.4 and in Figure 5.

The present study found a strong positive (r2 = 0.43, p-value < 0.001) correlation between the allophane concentration and the pH value and and a strong negative (r2 = 0.77, p-value < 0.001) correlation between the (Alpyr+Fepyr):(Alox+Feox) ratio and the pH value, respectively (Figure 5; A, B).

**Comment 28:**

Line 29, page 13: what do you mean with "continuous"? That value is also not normalized by time, I can't understand that adjective in such context. To me "continuous" could refer here to a rate of inputs that did not change over 15 and over 50 years, but this is a (cumulative, so integrated over time and not a rate) mass. And what that increase the same for all the

15   stands?!? What do you mean 15 t C ha-1 between 15 and 50 years?

**Author's answer:**

The sentence was corrected.

20   **Revised version:**

Nevertheless, afforestation with mountain birch leads to a significant (p = 0.05) increase of the SOC stock (+15 t C ha-1; +48 %) for birch stands between 15 and 50 years.

25   **Comment 29:**

Line 14-15, page 13: you wanted to "evaluate the SOC sequestration potential of afforestation on severely degraded soils in southern Iceland." and your key message is a recommendation about caution in choosing the sampling depth for soil surveys?!? I think you should focus a bit more on your main aims, you have some information there. And try to be consistent with such aims, write down your hypotheses, test them (also statistically) and tell me more about how it went. I wouldn't use

30   the last line for a recommendation that just points out some shortcomings of your study, actually.

**Author's answer:**

We improved the conclusion chapter by adding more essential results and place three key messages which are the findings of the study.

**Revised version:**

The study aimed to evaluate the SOC sequestration potential of afforestation on severely degraded soils in southern Iceland. For this, we measured the SOC stocks of differently-aged afforested birch stands and compared them with those of eroded and degraded soils, re-vegetated grasslands and non-degraded woodlands which have escaped the soil erosion, respectively. In addition, the SOC quality of all sites was analyzed by physical soil fractionation. The present study differentiated between the physically separated SOC pools, which allowed for the evaluation of the success of afforestation by mountain birch on a landscape with highly diverse soil patterns and SOC distributions.

The results of the present study also clearly show that undertaking research on soil organic carbon patterns on severely degraded soils within this area is challenging, owing to the high SOC stocks (0-30 cm) of these degraded soils. Nevertheless, afforestation with mountain birch leads to a significant ($p = 0.05$) increase of the SOC stock (+15 t C ha-1; +48 %) for birch stands between 15 and 50 years. Afforested birch stands can still potentially accumulate SOC after 50 years of growth, due to their significantly ($p = 0.05$) lower SOC stock (+13 t C ha-1; +28 %) compared to naturally, old growth birch woodlands. During this time, the POM mass (+131 mg g-1 soil; +300 %) and POM-C concentrations (+35 mg g-1 soil; +285 %) increase during the succession of the mountain birch ecosystem. These increases were mainly observed in the top 10 cm of the mineral soil. Further, at least 56 % of the total SOC stock (0-30 cm) was found in the HF- and '< 63 μm' fractions and at all tested sites most of the carbon was stored in the < 63 μm fraction. Even severely degraded soils contain considerable amounts of the SOC stocks. Due to the increased amount of POM-C stock and the doubling of the DOC stock, it, however,seems that afforestation leads to SOC pools which are more vulnerable to release C to the atmosphere. The first key message is that severely degraded, un-vegetated soils can sequester considerable amounts of SOC and there is still a potential of SOC sequestration after 50 years of plant growth. Second, the standardized soil sampling depth of 30 cm needs to be vertically subdivided for evaluating the success of restoration regarding SOC sequestration on severely degraded soils. Third, the interaction of the organic material with the mineral phase of such volcanic soils needs to be studied in more detail. . Regarding the chosen setup approach, the applied space-for-time substitution approach showed limited success by reason of the heterogeneity of the parent material and its SOC properties at greater soil depths, In such cases, it would be more effective to use permanent plots and a long-term monitoring approaches to assess soil development during vegetation restoration, as initially suggested by Johnson and Miyanishi (2008), carried out by Arnalds et al., (2013) and Thorsson (in prep.), and further developed by Bárcena et al., (2014). Hence, the fourth key message of the study is that the establishment of chronosequence plots on severely degraded soils needs to be applied with caution.

Basel, 22 April 2019, M. Hunziker

Dear Robert Qualls

we want to thank you for your valuable review. Some of your remarks concerning the statistics and the topic of the essential status t0 for chronosequence studies were already key points which we also discussed during the data analysis process and

5 the writing of this manuscript. Due to the small sub-datasets per age and depth class and the not-normal distribution of these sub-datasets, we decided to apply the Wilcoxon rank-sum test (WRS), which is applied if the statistical requirements for T-test are not given. Further responses on this topic are listed below.

At this point I also apologize that we have not earlier answered to your reviews. The reason is that this manuscript records

10 one part of my PhD studies and I, as main author of the manuscript, have not been employed at any research institute for more than 1.5 years. Since that time, I work at an enterprise in the private industry. This is not an excuse to ignore your review comments, but the time and software recourses are very limited or no longer available. Nevertheless, we modified our manuscript according to your comments as thoroughly as possible, constructed replies to your comments and submit the revised manuscript and the answer of the authors hereby.

Kind regards

Matthias Hunziker, main author.

Referee 2; Robert Qualls

**General comment**

This manuscript describes a very interesting study of the accumulation of carbon, particle density fractions and the clay
fraction that would be relevant to adsorption of carbon in volcanic soils. It would be relevant to the literature on soil
development during primary succession on volcanic soils, and perhaps to secondary succession on volcanic soils. One thing
that is unique is that unlike in many studies of soil development during succession, there is only one species of tree involved,
with one "variable" removed (with the exception of the grassland which provides and interesting contrast with deposition of
carbon at different depths.

As the authors acknowledge, there is unfortunately no "time zero" for the afforestation of the birch since the barren plots
seem to have organic matter left from a previous era when it must have been vegetated, as indicated by C contents that are
greater, even at depth than the young birch plots. Perhaps some initial state can be inferred by extrapolation to zero time in
the birch time sequence.

The methods used were very pertinent to a study of soil development on volcanic substrates. The analyses of allophane and
Fe and Al oxyhydroxides are just what this reviewer used in comparable studies. The separation of carbon by density
fractions are also what Sollins et al. (see reference below) recommended to monitor the deposition of root detritus vs. the
adsorbed or occluded carbon that might be expected with allophane and Fe/Al oxyhydroxides interactions.

**Author's answer**

An exponential function based on the time-dependent SOC stocks (0-30 cm) of B15-B50 as input data (computed in Excel)
showed an SOC stock as initial status (t0) of 26.25 t C ha-1 ($y=26.246e0.0111x$, $R2=0.44$). This is a quite smaller SOC stock
value than found at Barren Land (39 t C ha-1). According to these, it seems that at the sites of B15, B20, B25 and B50 the
initial SOC stock before any afforestation activities starts is distinct lower than the used initial status of severely degraded
land (Barren Land) in the present study.

According to the comment of the editor, this idea was not further followed up.

**Revised version**

No changes were made.

**Comment 1**

There are a few things that I might suggest could be made clearer to the readers. In the description of the history of the sites,
I was not able to follow which plots actually used for the study were associated with each history. Perhaps it would help to

have a table listing each group of plots (barren, planted birch, natural birch, grassland) and relevant elements of history (previous land use, eroded, volcanic desert, volcanic sand deposition, etc.). In many comparable studies of chronosquences, a key question is the degree to which all vegetation/age types originated from the same parent material. Obviously they are all of volcanic origin, but some had different histories and there is no true "initial state" since there appears to be a buried A horizon.

**Author's answer:**

In our opinion, the description is good enough and a table would overload the section which already contains Figure 1 about the location and the setup of the soil sampling. In the revised version, we labeled the different tested categories.

**Revised version:**

No changes were made

**Comment 2**

Perhaps clarify the discussion as to which sites can be considered subsets of "vegetation/age" classes can be considered as having the same initial states that differ by age or vegetation.

**Author's answer:**

In the discussion, we considered this comment.

**Revised version:**

The results indicate that spatial variability must be taken into account when analyzing SOC of volcanic soils, especially when deeper than 10 cm, between the sampled sites and the land cover categories (i.e. grassland, barren, etc). This is even more relevant in landscapes with past or recent erosion processes as soil forming process. Thus, the equality or comparability of the sites, except for the studied variable, is not ensured for space-for-time substitution sampling approaches under such circumstances as performed in the present study (Walker et al., 2010). Hence, it is misleading to use the selected Barren Land sites, which were selected at 4 km distance from the afforested sites (Birch15, Birch20, Birch25 and Birch50) and 15 km from Birchnat, as initial status (t0) for discussing the effect of afforestation and calculating any SOC sequestration rates.

**Comment 3**

Study design and replication. The following paragraph makes it difficult to figure out the experimental design and replication: "Each of the land cover types and age categories described above was represented by three test sites, resulting in

a total of 21 sampling sites (Figure 1; E). . . .. . .. . .. . .. ..r. At each site, five soil pits were randomly placed. At the woody sites, sampling occurred within one half of the crown diameter of a dominant mountain birch (Betula pubescens Ehrh. ssp. czerepanovii) tree. The soil was sampled with a cylindric metal core (Eijkelkamp Soil & Water, Giesbeek) of 100 cm3 volume and 5 cm in diameter at given soil intervals (0- 5, 5-10, 10-20 and 20-30 cm). The five subsamples per depth interval were immediately mixed in order to form one composite sample. Thus, each depth interval per category was represented by three composite samples (Figure 1), resulting in a total of 84 composite samples."

It is difficult to figure out the experimental design from paragraph and figure (Figure 1) seems to have some contradictions. There were 5 pits in each site. Part of the problem is the use of the words "land cover types" and "sites". Many authors use "site" to indicate the "treatment" and "plot" as the unit that serves as a replicate. I realized these were not randomly allocated treatments, but the nomenclature is confusing making it difficult to tell that there are 3 replicates per "vegetation/age" class. What is "category" in "depth interval per category, is this the same as site? Could site be referred to as "plot"?

**Author's answer:**

We see the problem which is mentioned by the reviewer. During the writing of the manuscript we intensively thought about the most appropriate terminology. Throughout the manuscript, we keep the terminology constant.

It is correct that there were 5 pits per site.

"Category" in "depth interval per category" is land cover category in combination with the age of vegetation growth e.g. "Barren Land", "Grass50, "Birch15". And the term "category" is not the same as "site" because we tested three sites per category.

In our study setup the term "site" is referred to as "plot" which serves as a replicate according to the reviewer.

In section 2.1 we made minor changes for a better understanding of the study setup.

**Revised version:**

Each of the land cover types (e.g. Barren Land, afforested birch stands) and age categories (e.g. 15, 20, 50 yrs old birch stand) described above was represented by three test sites (3 replicates) (Figure 1; E).

Thus, each depth interval per category was represented by three composite samples (3 replicates per depth interval) (Figure 1), resulting in a dataset of total of 84 composite samples.

**Comment 4**

In Figure 1, the map is useful. But, in the maps B, C, and D I do not see asterisks, triangles, etc. as it says in the caption.

The list of sites, profiles, and composite samples is only confusing. Perhaps you could list "vegetation/age" classes, "number of plots or sites within each class", "subsamples composited within each plot". . . to make the number of true replicates apparent.

5 **Author's answer:**

The points of the test sites were categorized as it is mentioned in the caption. However, we keep the list with the numbers of test sites, soil pits, collected samples and composite samples.

The capture of the Figure was changed.

10 **Revised version:**

Figure 1 was changed.

Figure 1: The topological map (equidistance = 100 m) showing the study area between the Ytri-Rangá River, Mount Burfell, Mount Hekla and Gunnarsholt (crossed cycle) in the south of Iceland (A). The locations of the naturally growing birch
15 woodland (B; asterisks) and the afforested (C; B15: circles, B20: triangles; B25: pentagons; B50: diamonds) and degraded (crosses) as well the revegetated (stars) test sites (D) are shown in more detail. The sampling scheme illustrates the age and vegetation characteristics of the different study sites and the applied soil sampling setup (E).

**Comment 5**

In the discussion, there are a couple of very relevant references that are comparable in terms of (1) the rate of carbon accumulation over time on volcanic soils, (2) the development of allophane and iron and aluminum oxyhydroxides and the role of adsorption of carbon, and (3) the use of density fractionation to examine the role of association of C with volcanic minerals and its refractory nature.

**Author's answer:**

We read the listed publications and cited Lilienfein et al. 2003 and Sollins et al. 1983.

**Specific comments:**

**Comment 6**

Abstract lines 26 through 29. The cause and effect does not seem clear. Suggested revision: "After 50 years of birch growth, the SOC stock is lower than that of a naturally growing birch woodland. Suggesting that afforested stands could sequester additional SOC beyond 50 years of growth."

**Author's answer:**

Reviewer's suggestion was accepted.

**Revised version:**

Another finding of the present study is that after 50 years of birch growth, the SOC stock was still significantly ($p = 0.05$) lower than that of the old-growth woodlands of Birchnat ($\Delta$ 15 t ha-1) (Figure 2). This means that the soils at Birch50 can sequester additional organic carbon during the succession towards mature woodlands which reflects the equilibrium state.

**Comment 7**

please spell out sodium polytungstate

**Author's answer:**

The suggestion was accepted and the sentence was changed.

**Revised version:**

[revised manuscript text omitted]

No. of test sites: 21
No. of soil profiles: 105
No. of collected soil samples: 420
No. of composite samples: 84

0-5 cm
5-10 cm
10-20 cm
20-30 cm

Barren Land

Initial

Birch$_{15}$  Birch$_{20}$  Birch$_{25}$  Birch$_{50}$

Afforested

Grass$_{50}$

Revegetated

Birch$_{nat}$

Undisturbed

[Figure]

**Figure 1: The topological map (equidistance = 100 m) showing the study area between the Ytri-Rangá River, Mount Burfell, Mount Hekla and Gunnarsholt (crossed cycle) in the south of Iceland (A). The locations of the naturally growing birch woodland (B; asterisks) and the afforested (C; B15: circles, B20: triangles; B25: pentagons; B50: diamonds) and degraded (crosses) as well the revegetated (stars) test sites (D) are shown in more detail. The sampling scheme illustrates the age and vegetation characteristics of the different study sites and the applied soil sampling setup (E).**

[Figure]

Figure 2: Median soil organic carbon stocks [t C ha$^{-1}$] in the mineral soil of the studied eroded (Barren Land), reclaimed (Grass50, Birch15, Birch20, Birch25 and Birch50) and old-growth (Birchnat) sites. The range of the error bars is shows the minimum and maximum values. The different shadings indicate the four sampling depths (0-5cm: diagonal lines; 5-10cm: rectangular squares; 10-20cm: b,w squares; 20-30cm: vertical lines). Within a sampling depth, significant differences (Mann-Whitney U Test, $p \le 0.05$) between the age classes are indicated by different letters. Further, significant differences (Mann-Whitney U Test, $p \le 0.05$) between the total studied soil depth (0-30cm) are shown above the stacked columns.

[Figure]

**Figure 3: SOC concentration [mg g$^{-1}$] of the fraction POM (A, E, I, M), HF (B, F, J, N), '< 63 µm' (C, G, K, O) and DOC (D, H, L, P) divided into the sampled soil depths (0-5, 5-10, 10-20 and 20-30 cm) for the reclaimed (Birch15, Birch20, Birch25, Birch50 and Grass50), eroded (Barren Land) and old-growth (Birchnat) sites. The boxes are show the minimum, median and maximum values. Note the variable scale of the Y-axis.**

[Figure]

**Figure 4: Cumulated carbon concentrations [mg g⁻¹] (0-30cm) within the analyzed SOC fractions for the reclaimed (Birch15, Birch20, Birch25, Birch50 and Grass50), eroded (Barren Land) and old-growth (Birchnat) sites. The boxes show the minimum, median and maximum values. Note the variable scale of the Y-axis.**

[Figure]

[Figure]

**Figure 5: Relationship between common properties of volcanic soils. The charts show the allophane concentration [%] as a function of pH value (H₂O) [-] (A),  unfractionated SOC concentration [%] (B) and '< 63 µm' SOC concentration [%] (C), as well as the amount of Al and Fe, in the form of organo-mineral complexes ((Fe$_{pyr}$+Al$_{pyr}$):(Fe$_{ox}$+Al$_{ox}$) molar ratio [-]), as a function of pH value (H₂O) [-] (D),  unfractionated SOC concentration [%] (E) and '< 63 µm' SOC concentration [%] (F). The observations (N = 84) are labelled based on the vegetation types: Barren Land (✳), Birch15 (▲), Birch20 (●), Birch25 (◆), Birch50 (■), Grass50 (+) and Birchnat (×). The dotted circles show all samples of Birchnat (0-5 cm, 5-10 cm), all samples of Birch50 (0-5 cm) and one sample of Birch25 (0-5 cm).**

---

## Editor Decision (ED1)

This study aimed to evaluate the SOC sequestration potential of afforestation on severely degraded soils in southern Iceland due to the forecasted high potential of these soils. ~~For this, we measured the SOC stocks of differently-aged afforested birch stands and compared them with those of eroded and degraded soils, re-vegetated grasslands and non-degraded woodlands which have escaped the soil erosion, respectively. In addition, the SOC quality of all sites was analyzed through physical soil fractionation. The present study differentiated between the physically separated SOC pools, which allowed evaluating the success of afforestation by mountain birch on a landscape with highly diverse soil patterns and SOC distributions.~~

Afforestation with mountain birch leads to an  increase of the SOC stock (0-30 cm) between the age of 15 and 50 years.  Since  the 50-year birch stands still contained   lower SOC stock than naturally, old growth birch woodlands. it appears  that the SOC stock equilibrium is not reached yet . Consequently, afforestation with the native mountain birch species is a successful strategy to sequester atmospheric carbon in  severely degraded volcanic soils by about 20 t C ha$^{-1}$. However, stored C is likely relatively labile with a disproportional rise in the POM fraction SOC (> 63 µm, < 1.8 g cm-3) compared to mineral-associated OC stored in the HF and '< 63 µm' fractions, especially in the top 10 cm. Indeed, the proportion of the latter SOC fraction declined to just more than half of the 0-30cm SOC stock in the afforested plots as opposed to over 90% in un-vegetated soils. As a consequence much of the newly stored C may not be sequestrated at all but is probably prone to loss again in the event of future change in OM inputs. Our approach thus reveals that detailed measurements on the SOC quality are equally needed to appreciate the SOC sequestration potential of restoration activities on severely degraded volcanic soils, rather than only measuring SOC stocks. Lastly, we found that severely degraded volcanic soils are surprisingly variable in their SOC stocks, with often inverse SOC profiles resulting from an interplay between soil erosion and burial by ash from volcanic eruptions. This highly local occurrence of specific SOC depth profiles even more so than normal necessitates a depth differentiated approach to deduce SOC storage resulting from land-use changes.

~~Only measuring the commonly used unfractionated SOC socks can therefore lead to misinterpretation of the sequestration potential of these soils. The study clearly shows that un-vegetated soils can contain certain amounts of SOC before afforestation activities begins. This, it is difficult to use un-vegetated sites and its SOC stocks as initial status before restoration activities begin. Under such conditions, it is advisable to use a depth resoluted sampling approach as well as a physical fractionation technique to extract the SOC deriving from the afforestation process. These fractionation analyses revealed that at least 56 % of the total SOC stock (0-30 cm) is stored in the HF and '< 63 µm' fractions. At the un-vegetated soils, this ratio is even higher than 90 %. And during the establishment of bush vegetation, the ratio of the concentration, the mass and the SOC stock in the POM fraction (> 63 µm, < 1.8 g cm-3) significantly increases, especially in the top 10 cm. Thus, the SOC change deriving from the afforestation effect can easily be disturbed by organic carbon originating from past vegetation which is found in the '< 63 µm' fraction and by sampling the top 30 cm. Instead of applying the space-for-time substitution approach (e.g. chronosequences), we therefore suggest investing more effort in depth-resoluted and qualitative SOC analyses or using permanent plots or a long-term monitoring approaches to assess soil development during vegetation restoration.~~

**Commented [SS1]:** Does not belong in a conclusions section.

**Commented [SS2]:** I would mention this, but more explanation was needed -> see the above track-changes version

**Commented [SS3]:** Awkward term: better 'depth-differentiated'

**Commented [SS4]:** No implications were connected to all of this. In the version above a link is made to proneness to loss of C after any future change in C-input

**Commented [SS5]:** Not a very astonishing conclusion

---

## Author Response (AR2)

Dear Steven Sleutel (Topical Editor)

we want to thank you for your valuable comments. We went through them. We finally reorganized and rewrote the conclusion as it was suggested by you. Also the minor corrections were done during this revision process.

Kind regards

Matthias Hunziker, main author.

Overall, the authors have well taken action according to points raised by both referees. There are, however, still one major and some minor points of attention requiring the submission of a revised manuscript, enlisted here below.

**Comment 1:**

Remove comma after 'et al' when the literature reference is part of the sentence: needs to be done corrected on many occasions throughout the manuscript.

**Author's answer:**

All comma after « et al. » within the sentences were deleted.

**Revised version:**

Blakemore et al. (1987), Zimmermann et al. (2007), Ellert at al. (2008), Rodeghiero et al. (2009), Guidi et al. (2014), Lilienfein et al. (2003), Hunziker et al. (2017), Snorrason et al. (2002), Vilmundardottir et al. (2015), Aradottir et al. (2000), Strachan et al. (1998), Barré et al. (2010), Barcena et al. (2014), Sollins et al. (1983), Arnalds et al. (2013).

**Comment 2**

When presenting correlations give r, not $r^2$

**Author's answer:**

In the revised version we use « r » as correlation parameter.

**Revised version:**

The present study found a strong positive (r = 0.68, p-value < 0.001) correlation between the allophane concentration and the pH value and a strong negative (r = -0.77, p-value < 0.001) correlation between the $(Al_{pyr}+Fe_{pyr}):(Al_{ox}+Fe_{ox})$ ratio and the pH value, respectively (Fig. 5; A, B).

**Comment 3**

Fig. instead of Figure – change everywhere

**Author's answer**

We changed it everywhere.

**Comment 4**

P9L13 'report' instead of 'reported' + change last part of sentence into 'report the initial SOC stock before the ecosystem change began'.

**Author's answer**

We changed the word to «report» and changed the last part of the sentence as it was suggested.

**Revised version**

However, Snorrason et al. (2002) and Ritter (2007) did not report the initial SOC stock before the ecosystem change began.

**Comment 5**

P10L28 The newly added part 'Such pedologic conditions can ask for soil C decay studies, as it was introduced by Barré et al., (2010). However, this was not the objective of the present study' is not clear with additional information. I suggest to again remove this reference to bare fallow plot studies – I do not see a straightforward link with this study.

**Author's answer**

We agree and deleted the two mentioned sentences.

**Revised version**

This implies that the soils of Barren Land contain a certain amount of SOC due to earlier soil formation processes prior to disturbance and SOC accumulation, and which occurred before the soil profile was truncated by soil erosion processes. Nonetheless, the SOC stocks (0-30 cm) of Barren Land are significantly ($p = 0.05$) lower than in soils under well-established and non-degraded ecosystems (Birchnat) (Fig. 2).

**Comment 6**

P10 '3.3.1 Vertical resolution of unfractionated SOC stocks' reads awkwardly, remove 'unfractionated'

**Author's answer:**

We agree and changed the title to « Vertical resolution of SOC stocks »

**Revised version:**

3.3.1 Vertical resolution of SOC stocks

**Comment 7**

The present Conclusion section is really a summary of the present study with even report of p-values of statistical tests etc. This section is not to the point and strongly repeats parts of the introduction and discussion. For example: "Nevertheless, afforestation with mountain birch leads to a significant (p = 0.05) increase of the SOC stock (+15 t C ha-1; +48 %) for birch 25 stands between 15 and 50 years. During this time, the POM mass (+131 mg g-1 soil; +300 %) and POM-C concentrations (+35 mg g-1 soil; +285 %) increase during the succession of the mountain birch ecosystem. These increases were mainly observed in the top 10 cm of the mineral soil. Further, at least 56 % of the total SOC stock (0-30 cm) was found in the HF- and '< 63 µm' fractions and at all tested categories most of the carbon was stored in the < 63 µm fraction."

This whole part mainly just reports results once more, with no conclusion drawn on key messages, implications for management, new insight etc. Other parts as well are not generic enough and too focused on the conducted study. This entire conclusion needs to be rewritten:

-The first 6 lines of the conclusion are a repetition of preceding parts and need to go - ' The results of the present study also clearly show that undertaking research on soil organic carbon patterns on severely degraded soils within this area is challenging, owing to the high SOC stocks (0-30 cm) of these degraded soils.' is not a crucial nor generic conclusion and rather – if anywhere – belongs in the Results & Discussion

**Author's answer:**

We rewrote the conclusion again.

**Revised version:**

The study aimed to evaluate the SOC sequestration potential of afforestation on severely degraded soils in southern Iceland due to the forecasted high potential of these soils. For this, we measured the SOC stocks of differently-aged afforested birch stands and compared them with those of eroded and degraded soils, re-vegetated grasslands and non-degraded woodlands which have escaped the soil erosion, respectively. In addition, the SOC quality of all sites was analyzed through physical soil fractionation. The present study differentiated between the physically separated SOC pools, which allowed evaluating the success of afforestation by mountain birch on a landscape with highly diverse soil patterns and SOC distributions.

Afforestation with mountain birch leads to a significant increase of the SOC stock (0-30 cm) between the age of 15 and 50 years. In addition after 50 years of birch establishment, birch stands contain still a significant lower SOC stock than naturally, old growth birch woodlands. This means that the SOC stock equilibrium is not reached after 50 years. Consequently, afforestation with the native mountain birch species is a successful strategy to sequester atmospheric carbon in the mineral phases of severely degraded volcanic soils by about 20 t C ha$^{-1}$.

The tested severely degraded volcanic soils showed an unexpected heterogeneity, such as the SOC properties due to landscape and soil development. Only measuring the commonly used unfractionated SOC socks can therefore lead to misinterpretation of the sequestration potential of these soils. The study clearly shows that un-vegetated soils can contain certain amounts of SOC before afforestation activities begins. This, it is difficult to use un-vegetated sites and its SOC stocks as initial status before restoration activities begin. Under such conditions, it is advisable to use a depth-resolved sampling approach as well as a physical fractionation technique to extract the SOC deriving from the afforestation process. These fractionation analyses revealed that at least 56 % of the total SOC stock (0-30 cm) is stored in the HF and '< 63 μm' fractions. At the un-vegetated soils, this ratio is even higher than 90 %. And during the establishment of bush vegetation, the ratio of the concentration, the mass and the SOC stock in the POM fraction (> 63 μm, < 1.8 g cm$^{-3}$) significantly increases, especially in the top 10 cm. Thus, the SOC change deriving from the afforestation effect can easily be disturbed by organic carbon originating from past vegetation which is found in the '< 63 μm' fraction and by sampling the top 30 cm. Instead of applying the space-for-time substitution approach (e.g. chronosequences), we therefore suggest investing more effort in depth-resolved and qualitative SOC analyses or using permanent plots or a long-term monitoring approaches to assess soil development during vegetation restoration.

**Comment 8**

Only form p15L29 more or less a more appropriate style is used -> use this as a guideline but severely un-vegetated soils may on the other hand need to be specified a bit more

Check carefully also for typos with spaces commas and periods in the final part of the conclusion

**Author's answer:**

We checked the manuscript again and corrected typos with spaces etc. and made also the following changes.

**Revised version:**

P2L6

[revised manuscript text omitted]

---

## Author Response (AR3)

Dear Steven Sleutel (Topical Editor)

we want to thank you for the reorganisation and valuable improvement of the conclusion. We agree and updated the manuscript by the suggested conclusion. We did not make any other changes.

Kind regards
Matthias Hunziker, main author.

Basel, 2nd June 2019

[revised manuscript text omitted]